# Improving Summarization with Human Edits

**Zonghai Yao** †
University of Massachusetts, Amherst
zonghaiyao@umass.edu

**Benjamin J Schloss**
Abridge AI Inc.
ben.j.schloss@gmail.com

**Sai P. Selvaraj**
Abridge AI Inc.
aps.prabhakar@gmail.com

## Abstract

Recent work has shown the promise of learning with human feedback paradigms to produce human-determined high-quality text. Existing works use human feedback to train large language models (LLMs) in general domain abstractive summarization and have obtained summary quality exceeding traditional likelihood training. In this paper, we focus on a less explored form of human feedback – Human Edits. We propose Sequence Alignment (un)Likelihood Training (SALT), a novel technique to use both the human-edited and model-generated data together in the training loop. In addition, we demonstrate simulating Human Edits with ground truth summaries coming from existing training data – Imitation edits, along with the model-generated summaries obtained after the training, to reduce the need for expensive human-edit data. In our experiments, we extend human feedback exploration from general domain summarization to medical domain summarization. Our results[1] demonstrate the effectiveness of SALT in improving the summary quality with Human and Imitation Edits. Through additional experiments, we show that SALT outperforms the conventional RLHF method (designed for human preferences) – DPO, when applied to human-edit data. We hope the evidence in our paper prompts researchers to explore, collect and better use different human feedback approaches scalably.

## 1 Introduction

Large-scale language model pretraining has become increasingly prevalent to achieve high performance on various natural language processing (NLP) tasks (Brown et al., 2020; Sanh et al., 2021; Chowdhery et al., 2022; Longpre et al., 2023; OpenAI, 2023; Cai et al., 2023). When applying these

models to a specific task, they are usually fine-tuned to maximize the likelihood of human-written text. While this strategy has led to markedly improved performance in many metrics, models still cannot consistently produce human-determined high-quality output. The NLP community has pointed out some key drawbacks of traditional fine-tuning. First, important errors (e.g. hallucinations) and unimportant errors (e.g. minor grammar errors) equally contribute to the final loss. Second, the model weighs the loss equally on all labeled data of different types, qualities, and difficulties. Third, distribution shifts in new data degrade performance (catastrophic forgetting)(Kirkpatrick et al., 2017).

| CC | Ground truth summary | |
|---|---|---|
| DR: Plus ribavirin, roughly based on your weight. Like 3 pills in the morning, 3 pills in the evening. | Start ribavirin 3 pills twice a day | |
| CCUser | $S_{AI}$ | $S_E$ |
| DR: Uh, and have you had any more chest pain? PT: I did, yeah, I do. | chest pain | Confirms chest pain. |
| DR: Uh, and have you had any more chest pain? PT: Not really. No. | chest pain | Denies chest pain. |
| DR: And then I have gemfibrozil 600 mg twice a day. DR: Fish oil, you do 2 capsules twice a day. | Fish oil. | Fish oil 2 capsules twice a day. |

Table 1: Example of conversation-to-notes summarization data from Clinician Conversations (CC) dataset and corresponding human-edit dataset, CCUser, where user-edited summaries– $S_E$, made from the AI-generated ones– $S_{AI}$, from our SOAP generation pipeline.

Some works tackle these problems with human feedback (HF). Specifically, they fine-tune language models with HF using reward learning (Stiennon et al., 2020; Ziegler et al., 2019). With a large amount of HF data, these works demonstrate that large-scale LMs, such as GPT-3 (Brown et al., 2020), have a text generation quality exceeding traditional likelihood training. However, the acquisition cost of large-scale HF is high, and whether smaller LMs can also benefit is not fully studied. In

---

[1]Code and the public dataset (Appendix A.2) is at https://github.com/seasonyao/LearnFromHumanEdit

† Work was done during internship at Abridge AI Inc

addition, because LLMs are often provided in the form of third-party APIs and are too large for many companies' and labs' infrastructure to host, smaller models (e.g., T5 family (Raffel et al., 2020)) still play important roles in many domains (e.g., medical), where privacy issues and pragmatic economics dominate decision-making strategies.

Our goal in this paper is to explore methods to train language models to improve the summary quality with HF inexpensively. HF for summarization can come in different forms. One is to obtain human scores for the summaries. Previous work (Stiennon et al., 2020; Ziegler et al., 2019) focuses on training a reward function through HF data and using such rewards as training objectives by comparing different summaries' scores. More recently, this is used by generative AI works (e.g., ChatGPT and GPT4 (Ouyang et al., 2022; OpenAI, 2023)), and they call the method RLHF. Another HF is obtaining edits to make the summary correct. The second approach is a natural way to collect feedback from users in workflows where users may be working off of an AI-generated summary in their workflow. For example, the summaries $S_E$, in Table 1 are the results of clinicians/scribes modifying our AI-generated EHR summaries $S_{AI}$. In addition, the second approach might be more data efficient in improving the summarization models than the first, as it conveys more granular information than a score for the entire summary. Human Edits from the second approach can also be converted to scores with simple rules like the percentage of edits, although this has not been studied extensively. Hence, from an ML data point of view, the second approach has certain unique advantages. Furthermore, large-scale expert feedback is hard to get using annotation ways in RLHF, considering the expert/user's time, cost, and willingness. But, Human Edits, which can be obtained from the users using the AI summaries for their work, may become a more reasonable alternative in various professional-knowledge-intensive domains.

We explore how to use Human Edits to improve summary quality. In addition to general domain summarization, we also focus on a medical domain summarization task in automatic clinical note generation from doctor-patient conversations, which is understudied due to privacy and data inaccessibility problems. Table 1 provides an example of a Clinician Conversation from our dataset (CC). We present our work from two experiments on a novel

technique, Sequence Alignment (un)Likelihood Training (SALT), which uses Human Edits and unlikelihood objectives together with the standard likelihood training paradigm to improve the summary quality. Unlikelihood training was proposed to reduce the probability of unlikely tokens predicted by models (Welleck et al., 2019).

In our first experiment, we use the Human Edits from physicians editing AI-generated clinical summaries from medical conversations to improve the summarization models. In our second, we explore how we can get similar benefits with pre-existing ground-truth human summaries that are not written as edits to the AI-generated summaries, which we call Imitation Edits. We refer to AI-generated summary $S_{AI}$, human-edit summary $S_E$, and imitation-edit summary $S_I$. We show how the unlikelihood objective can be generalized to improve the summary quality together with ($S_{AI}$, $S_E$) and ($S_{AI}$, $S_I$) pairs. In addition, our results show that SALT stably improves summary quality for T5 (small and large) summarization models with Human and Imitation Edits. Further experiments show how SALT can address the catastrophic forgetting problem arising from the distribution difference between $S_{AI}$ and $S_E$ with the help of RSALT, which is an improved version of the Replay-based methods in Continual Learning (Rebuffi et al., 2017).

Finally, to compare SALT and RLHF, we experiment with SALT and Direct Preference Optimization (DPO) (Rafailov et al., 2023) on human edit data and demonstrate the superiority of SALT on this type of human feedback.

To conserve space constraints, we have relegated specific contents to the appendix. In Appendix A.1 and A.2, we provide definitions of the SOAP Structure and implementation details. In Appendix A.3, we focus on the utilization of Imitation Edits and SALT for training on publicly available datasets, accompanied by the experimental results. Lastly, in Appendix A.4, we have more discussion about the relation between SALT and various other RLHFs.

**In summary, our contributions are as follows:**

- To our knowledge, we are the first to extend current HF trends in summarization research to the automatic clinical note-generation task.

- Different from the form of HF used in previous work, we explore Human Edits to improve summary quality in this paper.

- We show how to construct Imitation Edits to reduce the need for expensive HF data.

- We show SALT extends unlikelihood training into a general framework using sequence alignment and further combines SALT and Replay-based methods (Rebuffi et al., 2017) into RSALT for tackling catastrophic forgetting.

- Finally, we show that SALT achieves better performance than DPO on human-edit feedback.

## 2 Related Work

Most directly related to our work is research on automatic clinical note generation from doctor-patient conversations (Schloss and Konam, 2020; Ramprasad et al., 2023; Krishna et al., 2020; Abacha et al., 2023a; Ben Abacha et al., 2023; Yim et al., 2023; Wang et al., 2023), and the difference is that those works focus on training a summarization model with pre-labeled data, while we focus on using HF further to improve the summary quality of the trained models.

Previous work used HF to train summarization models with reinforcement learning (RL) (Böhm et al., 2019; Ziegler et al., 2019; Stiennon et al., 2020) and used GPT-2 and GPT-3 to optimize HF across various summarization tasks. These RL-based methods focus on training a reward function through HF data and use such rewards as training objectives by comparing different summaries (RLHF). Recently, some RLHF variants collect or use rewards more flexibly and stably (Akyürek et al., 2023; Dong et al., 2023; Zhao et al., 2023; Yuan et al., 2023). We introduce unlikelihood training as an additional learning objective in supervised learning. Our technique aims to decrease the probability of unlikely sequences, defined as those which appear in the $S_{AI}$ but not in $S_E$, and increase the probability of verified sequences, which are in $S_{AI}$ and reinforced by $S_E$, as well as novel sequences which do not appear in $S_{AI}$ but do appear in $S_E$.

Unlikelihood training (Welleck et al., 2019) involves adding unlikelihood loss to lower the probability of negative candidates. Previous work has explored many scenarios with various negative candidates for unlikelihood training, including: style transfer (Devaraj et al., 2021), repetition, copying, and contradictions (Li et al., 2019), factuality (Cao and Wang, 2021), text degeneration (Su et al., 2022), and clinical summarization (Adams et al., 2022). In this work, we align the $S_E$ with $S_{AI}$ to identify negative candidates and train different tokens with unlikelihood and likelihood loss. We also show that our experiments on Human Edits

| Subsection for S, O, A, P | CC | CCUser |
|---|---|---|
| Family Medical History | 9.155 | 9.131 |
| Past Surgical History | 6.070 | 6.957 |
| Review of Systems | 8.043 | 7.183 |
| Chief Complaint | 4.199 | 4.162 |
| Allergies | 6.000 | 7.523 |
| Past Medical History | 5.158 | 4.435 |
| Social History | 8.631 | 9.880 |
| Medications | 6.618 | 3.762 |
| Immunizations | 5.758 | 7.281 |
| Laboratory and Imaging Results | 8.352 | 8.544 |
| Assessment | 29.31 | 33.85 |
| Diagnostics and Appointments | 9.724 | 10.73 |
| Prescriptions and Therapeutics | 10.42 | 7.928 |

Table 2: Average words in CC and CCUser.

can be extended to Imitation Edits to reduce the need for HF data which can be expensive to get.

## 3 Dataset

### 3.1 Clinician Conversations (CC) Dataset

This dataset is a collection of 63000 consented doctor-patient de-identification conversations with human transcripts with an average duration of 9 minutes. We segmented the dataset to create training, validation, and test sets of 52,000, 5,000, and 6,000 files each while controlling important characteristics of the distribution in each split. The transcripts of the conversations were annotated according to the traditional SOAP format [2]. A SOAP note can contain numerous observations that are grounded to shorter excerpts from the transcript via timestamps that relate back to the original audio. There are several sections and subsections in the SOAP structure, each of which needs specific information and is written in a different format. Table 2 shows the average length span of different subsections is large.

### 3.2 CCUser Dataset

In order to generate SOAP notes from doctor-patient conversations, our pipeline follows (Ramprasad et al., 2023; Krishna et al., 2020). We first record the clinical conversation, then transcribe it either using humans or using Google's medical-conversations Automatic Speech Recognition (ASR) service. Then, using our proprietary models, we classify utterances into SOAP sections. Finally, using our section-conditioned summarization model trained on the CC dataset, we generate summaries for each of the utterance clusters belonging to each section.

---

[2]SOAP structure details can be found in the Appendix A.1.

We use our pipeline to extract SOAP summaries for our clinician users who record their conversations with their patients via a mobile app. The generated summaries were edited by scribes and doctors using our dashboard for their documentation tasks. The dashboard is built for doctors and scribes to check and fix AI-generated summaries in their regular workflow quickly. Hence, we didn't enforce any training/instructions that might make the data more useful for research, and the users were free to use the dashboard as they saw fit.

The distribution of the CCUser dataset differs from the CC dataset in the following ways. First, CC uses human-written transcripts as training inputs, while CCUser uses our pipeline's inputs from ASR transcripts rather than human-tagged utterances. Second, the average length of a conversation was 20 min for CCUser compared to 9 min for CC dataset, which could mean more complex conversations. The dataset has 215 ASR transcripts with AI-generated notes (along with the Human Edits) from 10 physicians. We randomly select 70 notes from 7 physicians as a training dataset, 10 for each physician, and divide the remaining 145 notes into evaluation and test sets. Finally, our dataset is split as a train:eval:test = 1279:1457:1458 – (utterance cluster, edited summary, AI summary) triplet.

## 4 Methods

Given a tokenized utterance cluster as input $U = [x_1, x_2, x_3, ...x_{lenU}]$, the CC summarization model $M$ generates a summary $S_{AI} = [y'_1, y'_2, y'_3, ...y'_{lenS_{AI}}]$ for it. The user edits this summary from $S_{AI}$ to $S_E$, where $S_E = [y_1, y_2, y_3, ...y_{lenS_E}]$. We aim to update parameters in $M$ based on both $S_{AI}$ and $S_E$. Let, $lenU$, $lenS_{AI}$, and $lenS_E$ be the number of tokens in $U$, $S_{AI}$, and $S_E$ respectively.

### 4.1 Sequence Alignment (un)Likelihood Training (SALT) using $S_{AI}$ and $S_E$

When a user edits a summary from $S_{AI}$ to $S_E$, they can modify or delete a span of tokens, insert a new span of tokens, or not change anything to a span of tokens. We want to use these Human Edits to improve our summarization models and produce outputs that are closer to the user's modified summary than before. We do this using both $S_{AI}$ and $S_E$ in the training. We train the model to:

(i) Lower the probability of producing words that the user deleted or modified in $S_{AI}$.

(ii) Reinforce the probability of producing words that the user didn't change in $S_{AI}$ and are retained in $S_E$.

(iii) Increase the probability of producing words that the new user added in $S_E$.

The loss functions to train the summarization model with $S_{AI}$ and $S_E$:

$$L_{S_{AI}} = \sum_{x \in S_{AI}} [\mathbb{1}_{AI-C}(t)\, w_{AI-C}\, L_p(x,t) + \mathbb{1}_{AI-NC}(t)\, w_{AI-NC}\, L_r(x,t)] \quad (1)$$

$$L_{S_E} = \sum_{x \in S_E} [\mathbb{1}_{E-C}(t)\, w_{E-C}\, L_p(x,t) + \mathbb{1}_{E-NC}(t)\, w_{E-NC}\, L_r(x,t)] \quad (2)$$

$$L_p(x,t) = -log\,(1 - p_\theta(x_t|x_{<t},\, U)) \quad (3)$$

$$L_r(x,t) = -log\, p_\theta(x_t|x_{<t},\, U) \quad (4)$$

Where:

1. $U$ is the utterance cluster used as input

2. $C$ and $NC$ mean "changed" and "not changed" tokens when we align $S_{AI}$ and $S_E$ sequences.

3. $\mathbb{1}_{AI-C}(t)$ and $\mathbb{1}_{AI-NC}(t)$ are the indicator function to signify if the token $x_t$ in $S_{AI}$ is changed or not-changed by the user. Similarly, $\mathbb{1}_{E-C}(t)$ and $\mathbb{1}_{E-NC}(t)$ corresponds to $S_E$.

4. $w_x$ are the loss weights, for example, $w_{AI-C}$ is the weight to penalize tokens that are in $S_{AI}$ but not in $S_E$.

5. $L_r(x,t)$ and $L_p(x,t)$ are the likelihood and unlikelihood loss functions

The losses $L_{S_{AI}}$ and $L_{S_E}$ used in the $(S_{AI}, S_E)$ pair are used to train the summarization model. The indicator functions used in the above equations can be found by tracking the user changes as they edit the summary or by aligning $S_E$ to $S_{AI}$ using a sequence alignment algorithm. We use sequence alignment (the Needleman-Wunsch Algorithm (Needleman and Wunsch, 1970)) in this work because our dashboard doesn't log the users' keystrokes. Assume we have a pair from $S_{AI}$ and the corresponding $S_E$, "patient takes one aspirin daily" and "patient doesn't want to take aspirin". We can align these two sentences as below:

| patient | – | – | – | takes | one | aspirin | daily |
|---------|-----|------|-----|-------|-----|---------|-------|
| patient | doesn't | want | to | take | – | aspirin | – |
| C | I | I | I | S | D | C | D |

Where "C" is "Correspondence" (matching), "I" is "Inserted", "D" is "Deleted", and "S" is "Substituted". Note that we do it on the token level in the implementation. For $S_{AI}$ word list ["patient", "takes", "one", "aspirin", "daily"], the corresponding indicator function in Equation 1 are:

$$\mathbb{1}_{AI-C}(t) = [0, 1, 1, 0, 1]$$
$$\mathbb{1}_{AI-NC}(t) = [1, 0, 0, 1, 0]$$

For $S_E$ word list ["patient", "doesn't", "want", "to", "take", "aspirin"], the corresponding indicator function in Equation 2 are:

$$\mathbb{1}_{E-C}(t) = [0, 1, 1, 1, 1, 0]$$
$$\mathbb{1}_{E-NC}(t) = [1, 0, 0, 0, 0, 1]$$

## 4.2 Imitation Edits

$S_E$ is a special kind of ground truth summary from the user. $S_E$ is obtained by the user using $U$ and $S_{AI} - S_E = Fn(U, S_{AI})$. An interesting question is whether we can approximate the edited summary $S_I$ (Imitation Edits), and use it to improve the models in the absence of actual Human Edits with SALT. In our work, we use the pre-existing ground-truth summaries as $S_I$ even though they were not explicitly written as edits to $S_{AI}$. Leveraging such data has several advantages. First, $S_E$ is not easy to obtain, approximating $S_E$ with $S_I$ can increase the amount of data available for unlikelihood training. And we will be able to use SALT even without human-edit data or any new annotations. Second, under the premise of ensuring that the Imitation Edits are of high quality, combining Human Edits and Imitation Edits can further improve the model's performance since both of them bring effective data points for training. Third, Imitation Edits can be used to solve the forgetting problem when we do SALT training with $S_{AI}$ and $S_E$, we show this in the next section.

To imitate Human Edits, we assume the original ground truth summary is generated from $S_{AI}$ and its utterance cluster $U$ (even though the ground truth notes were written independently). Similar to the above setting with $S_{AI}$ and $S_E$, we use the alignment algorithm to align $S_{AI}$ and $S_I$. Then we calculate $L_{S_I}$.

$$L_{S_I} = \sum_{x \in S_I} [\mathbb{1}_{I-C}(t) \, w_{I-C} L_p(x, t) + \mathbb{1}_{I-NC}(t) \, w_{I-NC} L_r(x, t)] \quad (5)$$

where $\mathbb{1}_{I-C}(t)$ and $\mathbb{1}_{I-NC}(t)$ signify if the token $x_t$ in $S_I$ is changed or not-changed compared to $S_{AI}$, and $w_x$ are the loss weights.

## 4.3 Replay-based SALT (RSALT) for Catastrophic Forgetting Problem

We continue training the model $M$ that has converged in the original summarization dataset (e.g., CC) on the Human Edits dataset (e.g., CCUser) to improve the summary quality, subjecting the model to the catastrophic forgetting problem because of the distribution differences between them. We use the traditional Replay-based methods, (Rebuffi et al., 2017), which sample a part of the data from the seen dataset (e.g., CC) and add it to the unseen data (e.g., CCUser), to address the catastrophic forgetting problem. Here, the likelihood loss is calculated for both sampled seen data $S_{I(seen)}$ and human-edit data $S_{E(unseen)}$ with the loss function $L = MLE_{S_{I(seen)}} + MLE_{S_{E(unseen)}}$, where we use Maximum Likelihood Estimation for the loss.

Following Section 4.1, we can use both $S_{AI(unseen)}$ and $S_{E(unseen)}$ to do SALT training. Following Section 4.2, for the sampled previously seen data, we can also get $(S_{AI(seen)}, S_{I(seen)})$ pairs and do SALT training. According to Equations 1, 2, 5, the loss function with RSALT is

$$L_{SALT} = L_{S_{AI(unseen)}} + L_{S_{E(unseen)}} \quad (6)$$
$$L_{RSALT} = L_{S_{AI(seen)}} + L_{S_{I(seen)}} \quad (7)$$
$$L = L_{SALT} + L_{RSALT} \quad (8)$$

# 5 Metrics

**ROUGE and UMLS-F1**  Models are evaluated with full-length F1-scores of ROUGE (Lin, 2004). We use QuickUMLS[3] to extract medical concepts from both model-generated and ground truth summaries and then calculate F1-scores for these two lists of concepts, which is named UMLS-F1 (Adams et al., 2023; Ramprasad et al., 2023).

**GPT4 & Human preference**  Recent work shows a higher correlation between human and GPT4 evaluation than traditional metrics (Moramarco et al., 2022; Gao et al., 2023; Fu et al., 2023), so we also use GPT4 preference as measurements to evaluate summary quality. Specifically, we instruct GPT4 to give preference ranking on different AI-generated summaries based on the conversation snippet and reference summary [4]. Similarly, we asked 2 medical students[5] to rate summaries from $CC$ based on the same information, for privacy reasons, we did not evaluate CCUser with humans. We discuss the Mean Reciprocal Rank (MRR) (Radev et al., 2002) of different models in Section 6.4. Generally, a higher MRR value implies that evaluators have more preference over an approach.

---

[3]https://github.com/Georgetown-IR-Lab/QuickUMLS
[4]Prompts can be found in Appendix.
[5]Both with hospital internship experience

**SAGE** ROUGE and UMLS-F1 measure the degree of "likelihood," i.e., they evaluate whether or not the model can generate something closer to some references. However, we don't just want to know how much "closer to $S_E$" is newly generated summary, but also how "far away from the bad part of $S_{AI}$" – spans that are changed by the Human Edits. To address this problem, we design an evaluation method to measure how likely machines are to make the same mistakes as before and how likely they are to generate summaries more like the target users (as identified during the editing process). We call this System output Against the Generated and Edited sentence (SAGE). Given the evaluation data $(U, S_{AI}, S_E)$, where $S_{AI}$ is generated by the model trained by the original summarization dataset (e.g., CC) and $S_E$ is edited by human based on $(U, S_{AI})$, we can get the new summary $S_{new}$ generated by the new model trained by Human Edits dataset (e.g., CCUser). Using $(S_{new}, S_{AI}, S_E)$, we can define three groups of words after removing stop words and punctuation in $S_{new}$:

1. $G_{w1(AI-E)} = \{w|w \in S_{AI} \wedge w \notin S_E\}$
2. $G_{w2(E-AI)} = \{w|w \notin S_{AI} \wedge w \in S_E\}$
3. $G_{w3(AI\cap E)} = \{w|w \in S_E \wedge w \in S_{AI}\}$

By training on HF, we aim to have $S_{new}$ closer to $S_E$ while avoiding the mistakes found in $S_{AI}$. So SAGE counts how many words in $S_{new}$ are in $G_{w1(AI-E)}$, $G_{w2(E-AI)}$, and $G_{w3(AI\cap E)}$. We call this word level SAGE ($SAGE_w$). Similarly, we can define $G_{c1(AI-E)}$, $G_{c2(E-AI)}$, $G_{c3(AI\cap E)}$ and make Concept-level SAGE ($SAGE_c$) based on UMLS concept overlap in $S_{new}$, $S_{AI}$, and $S_E$.

We have two assumptions regarding SAGE:

1. users can accept machines making some mistakes, but they can't tolerate machines making the same mistake, again and again.
2. users will be more satisfied if the model, over time, learns to generate outputs more similar to the user's edited summaries

According to Assumption 1 and 2, a model trained on HF should be able to generate less content belonging to $G_1$ ($G_{w1}$ and $G_{c1}$), and more content belonging to $G_2$ ($G_{w2}$ and $G_{c2}$). The model should also be able generate $G_3$ ($G_{w3}$ and $G_{c3}$) since $G_3$ represents human-verified information.

## 6 Experiments

We use the following symbols:

1. $M$ refers to models that are trained and al-

| | CCUser$_{eval}$ | | CC$_{eval}$ | |
|---|---|---|---|---|
| | R1 | U-f | R1 | U-f |
| $M$ | - | - | 36.07 | 48.97 |
| SALT$_l$ | 57.77 | 61.02 | 34.27 | 46.45 |
| SALT$_{l_d}$ | 57.70 | 61.06 | 34.46 | 46.58 |
| SALT$_{l_i}$ | 57.84 | 60.81 | 34.68 | 46.77 |
| SALT$_u$ | 57.57 | 61.09 | 34.47 | 46.64 |
| SALT$_{l+u}$ | 58.39 | 62.13 | 34.79 | 47.06 |
| SALT$_l$+RSALT$_l$ | 59.57 | 62.52 | 35.55 | 48.25 |
| SALT$_{l+u}$+RSALT$_l$ | 59.60 | 62.57 | 35.43 | 48.20 |
| SALT$_l$+RSALT$_{l+u}$ | 59.88 | 62.60 | 36.24 | 48.42 |
| SALT$_{l+u}$+RSALT$_{l+u}$ | 60.43 | 63.44 | 36.26 | 48.69 |

Table 3: Human Edits results. Compared to the likelihood training SALT$_l$, our proposed SALT$_{l+u}$ has better performance on both new human-edit CCUser$_{eval}$ and the model's prior training CC$_{eval}$ dataset, when using just CCUser$_{eval}$ for training (Section 6.1.1). Further, we show that the catastrophic forgetting problem can be addressed with Replay-based argumentation to our method– RSALT (Section 6.3). [6]

ready converged on the CC dataset. All methods below are initialized from $M$ and continue training on $S_E$, $S_I$, and $S_{AI}$.

2. SALT$_l$: the baseline, which is only based on likelihood training on $S_E$ or $S_I$
3. SALT$_{l_d}$ (or SALT$_{l_i}$): likelihood training on $S_E$ or $S_I$, but with decreased (or increased) weights for $\mathbb{1}_{E-C}$ or $\mathbb{1}_{I-C}$ tokens
4. SALT$_u$: only unlikelihood training on $S_{AI}$
5. SALT$_{l+u}$: both likelihood (on $S_E$ or $S_I$) and unlikelihood (on $S_{AI}$)
6. SALT$_x$: all the above SALT variations
7. SALT$_x$+RSALT$_l$ is the traditional replay-based method. When continuing to train $M$ with different SALT variations on new data, this method will sample a part of the data from the dataset that $M$ has already seen and use them for training with likelihood loss.
8. SALT$_x$+RSALT$_{l+u}$: following Section 4.3, RSALT treats sampled data from the replay-based method as imitation-edit data and uses both likelihood and unlikelihood training.

### 6.1 SALT in human-edit dataset

#### 6.1.1 Analyzing the behavior of SALT

In Table 3, the evaluation on the CCUser$_{eval}$ shows compared to the regular likelihood training

---

[6]There are no scores for $M$ on human-edit evaluation data (CCUser) here. Because human-edit data is directly modified from $M$'s $S_{AI}$, so it is unfair to calculate the scores of its $S_{AI}$ and human-edit data and compare with other methods.

| $\frac{SALT_x}{SALT_l}$ | $SAGE_w$ | | | $SAGE_c$ | | |
|---|---|---|---|---|---|---|
| | $G_{w1}\downarrow$ | $G_{w2}$ | $G_{w3}$ | $G_{c1}\downarrow$ | $G_{c2}$ | $G_{c3}$ |
| $SALT_l$ | 1 | 1 | 1 | 1 | 1 | 1 |
| $SALT_{l_d}$ | 0.982 | 0.889 | 1.005 | 1.022 | 1.011 | 1.001 |
| $SALT_{l_i}$ | 0.992 | 1.043 | 1.022 | 1.026 | 1.080 | 1.009 |
| $SALT_u$ | 0.833 | 0.824 | 0.977 | 0.894 | 0.954 | 0.981 |
| $SALT_{l+u}$ | 0.946 | 0.926 | 1.029 | 0.990 | 1.068 | 1.026 |

Table 4: Word-level and concept-level SAGE for CCUser$_{eval}$ normalize by $SALT_l$ as the baseline.

($SALT_l$), changing loss weights for $\mathbb{1}_{E-C}$ tokens in likelihood training ($SALT_{l_d}$ or $SALT_{l_i}$) can bring changes to their performance. Predictably we see in Table 4, that $SALT_{l_i}$ produces higher $G_{w2}$ than $SALT_{l_d}$, and the trends in other columns are not as pronounced since $S_{AI}$ isn't considered. Similarly, $SALT_u$ produces lower $G_{w1}$ than the others. However, $SALT_{l+u}$ achieves significantly higher performance on both CC and CCUser. We further show how we can manipulate a model's behaviors using different SALT through SAGE in Table 4.

First, $SALT_l$ only uses $S_E$, and all tokens in $S_E$ contribute to the loss equally. SALT can increase or decrease the emphasis of the model on $\mathbb{1}_{E-C}$ through different weights on the loss function. Increasing the loss weight of $\mathbb{1}_{E-C}$ will make the model generate more words/concepts belonging to $\mathbb{1}_{E-C}$ ($G_{w2}$ and $G_{c2}$), which follows our SAGE Assumption 2. While reducing the loss weight of $\mathbb{1}_{E-C}$ will make the model generate fewer words and concepts belonging to $\mathbb{1}_{E-C}$ ($G_{w2}$ and $G_{c2}$), at the same time it can also reduce the generation of words/concepts belonging to $\mathbb{1}_{AI-C}$ ($G_{w1}$ and $G_{c1}$), which satisfies our SAGE Assumption 1. So $SALT_{l_d}$ and $SALT_{l_i}$ make the model better for users according to the SAGE metric.

Second, unlike the three above SALT variations, $SALT_u$ only uses $S_{AI}$ but it knows which tokens in $S_{AI}$ belong to $\mathbb{1}_{AI-C}$ and $\mathbb{1}_{AI-NC}$ respectively. So $SALT_u$ significantly reduces the words and concepts belonging to $\mathbb{1}_{AI-C}$. However, because the data of $\mathbb{1}_{E-NC}$ has not been seen, $SALT_u$ rarely generates related words and concepts.

Finally, $SALT_{l+u}$ has more granular information– that tokens belonging to $\mathbb{1}_{AI-C}$, $\mathbb{1}_{AI-NC}$, $\mathbb{1}_{E-C}$, and $\mathbb{1}_{E-NC}$ in $S_{AI}$ ($S_E$) through their corresponding loss weights. Therefore, $SALT_{l+u}$ can learn the more suitable distribution, which decreases the generation of words and concepts belonging to $\mathbb{1}_{AI-C}$ while increasing the generation of words and concepts belonging to $\mathbb{1}_{AI-NC}$, $\mathbb{1}_{E-C}$ and $\mathbb{1}_{E-NC}$.

### 6.1.2 Reducing the forgetting problem

In Table 3, we see a dip in evaluation metrics for $SALT_l$ in the old evaluation dataset $CC_{eval}$ when we train the model trained on the CCUser – catastrophic forgetting. The reason could be the distribution difference between CCUser and CC dataset described in Section 3.2. Both $SALT_u$ and $SALT_{l+u}$ have different degrees of improvement in ROUGE-1 and UMLS-F1 on $CC_{eval}$ data. This result shows that SALT training also alleviates the forgetting problem to a certain extent.

One widely used and effective technique to reduce catastrophic forgetting is the replay-based method, which mixes in the seen data the model was trained on (e.g., CC). In this work, we set the ratio of CCUser and CC data to 2:1. That is, assuming that there are $n$ CCUser data, we will sample $0.5 * n$ CC data to train together [7]. Table 3 shows that $SALT_x$+$RSALT_l$ is effective in helping the model reduce the catastrophic forgetting problem. Adding the sampled seen data improves the model's performance in both the new – CCUser and the original – CC data. However, we still see a reduction in the performance of $SALT_x$+$RSALT_l$ in the CC dataset compared with $M$, which shows that the traditional replay-based method cannot completely solve this problem. In Section 6.3, we show how we address the problem further with SALT, imitation-edit data, and RSALT.

### 6.2 SALT in imitation-edit dataset

SALT uses the relationship between $S_E$ and $S_{AI}$ to get better performance than using just $S_E$ and likelihood training. In this section, we show that we can directly improve the summarization model $M$ using a similar relationship between $S_I$ (the ground truth data) and $S_{AI}$ without new human-edit data or additional annotation, i.e., by assuming that the $S_I$ is the output of human-edit data on $S_{AI}$. Simulating Human Edits this way lets us 1) demonstrate the effectiveness of SALT on a public dataset that does not have the human-edit component in them,[8] and 2) reduce the amount of Human Edits needed as it is hard to get.

Although both come from humans, $S_E$ and $S_I$ are fundamentally different in their relationship with $S_{AI}$. The former is modified from $S_{AI}$ while humans generate the latter from scratch. There-

---

[7] Adjusting this ratio will bring some improvements in certain $SALT_x$, but we found that 2:1 has relatively good performance in most $SALT_x$, so we use this ratio uniformly.

[8] Due to the space limit, we put results in the Appendix.

| | M | $l$ | $l_d$ | $l_i$ | $u$ | $l+u$ |
|---|---|---|---|---|---|---|
| R1 | 36.07 | 35.77 | 35.76 | 35.65 | 37.39 | 36.16 |
| U-f | 48.97 | 48.86 | 48.60 | 48.97 | 49.45 | 49.24 |

Table 5: SALT results for imitation-edit experiments. The imitation-edit data come from the training dataset which the model $M$ has already seen by assuming the ground truth is generated by editing the model's output.

| | $CC_{test-r}$ | | $CC_{eval}$ | |
|---|---|---|---|---|
| | R1 | U-f | R1 | U-f |
| M | 36.01 | 58.15 | 36.07 | 48.97 |
| $SALT_l$ | 36.09 | 57.55 | 36.14 | 48.50 |
| $SALT_{l+u}$ | 36.57 | 58.12 | 36.28 | 48.84 |
| $SALT_l+RSALT_{l+u}$ | 36.73 | 57.48 | 36.61 | 48.61 |
| $SALT_{l+u}+RSALT_{l+u}$ | 36.74 | 58.48 | 36.65 | 48.77 |

Table 6: Imitation Edits experiments. Here the imitation-edit data comes from a subset of the corresponding test dataset (we don't use them in the table for metrics), which $M$ has never seen before. We use CC-test for SALT and CC-train for RSALT during training.

fore, $S_E$ is directly dependent on $S_{AI}$, but $S_I$ is not. Consequently, even though $S_E$ and $S_I$ are dependent on the same data as input, the differences between $S_{AI}$ and $S_I$ are likely to be larger than between $S_{AI}$ and $S_E$. We can see this difference in the average percentage of changed tokens − $\mathbb{1}_{E-C}$ and $\mathbb{1}_{I-C}$ is 1, the former (6.17%) is much lower than the latter (45.59%). Hence, after we do sequence alignment between $S_I$ and $S_{AI}$, we perform a two-step post-processing operation [9] to ensure the training stability, which helps us to reduce the percentage of changed tokens from 45.59% to 19.07% with an acceptable amount of data lost (21.38%).

### 6.2.1 Imitation Edits using seen data

We use the training data from CC to experiment with the effects of SALT and Imitation Edits on seen data. First, for the CC dataset, the results in Table 5 show that continuing to use likelihood loss on the training dataset to train the already convergent $M$ does not improve the performance and leads to overfitting. However, when we use $S_I$ as imitation-edit data and do SALT training on it with $S_{AI}$, we can see an improvement. Second, we see similar results for the CNN dataset. Even though there is no performance degradation arising from overfitting for $SALT_l$, doing SALT training with $S_I$ and $S_{AI}$ can improve the performance more than using just the likelihood training. These results show that we can get additional improvement on the model by continuing to train it with SALT on

the seen dataset even if the model is already converged (on the seen/original training data). Third, different from previous human-edit results, $SALT_u$ of CC is better than $SALT_{l+u}$. We think this is because $M$ has started to overfit on CC data, so continuing to add likelihood to the original training data reduces the scores.

### 6.2.2 Imitation Edits using unseen data

We use a part of the test dataset (not used in the evaluation) from CC to experiment with the effects of SALT and Imitation Edits on unseen data. In Table 6, we take $M$ (trained on CC-train) and train it with a part of CC-test as the imitation-edit data with SALT. We take the remaining test data of the CC-test to evaluate the model performance in new imitation-edit data and then use CC-eval to evaluate the model performance in the original data. In imitation-edit evaluation results ($CC_{test-r}$) of Table 6, $SALT_{l+u}$ has better performance than the baseline method $SALT_l$, which is consistent with our results using human-edit data in Table 3. In the original data evaluation results ($CC_{eval}$) of Table 6, although there was no forgetting problem arising from distribution shift, $SALT_{l+u}$ still has a higher score than the baseline model $SALT_l$.

### 6.3 Solving forgetting problem with RSALT

Through previous analysis, we see that SALT helps $M$ to continue training on human-edit data or imitation-edit data. In Section 6.1.2 and 6.2.2, we observed that the traditional replay-based method cannot completely solve the catastrophic forgetting problem, so the performance of $SALT_x+RSALT_l$ on Table 3 and 6 is still lower than $M$'s performance if there are distribution differences.

We report the results of $SALT_x+RSALT_{l+u}$ in Table 3 and 6. We find that $SALT_x+RSALT_{l+u}$ does not have the forgetting problem when continuing to train with human-edit data. We attribute this result to the data augmentation that RSALT brings to the traditional replay-based method. RSALT not just reuses the seen data to prevent the model from forgetting the learned distribution but also uses the output generated by the model itself with SALT to expand the effective training data points further.

### 6.4 Preference Evaluation

In CC dataset, GPT4 (on 500 data points) ranks $SALT_{l+u}+RSALT_{l+u}$ higher than other variations ($SALT_l$ and $SALT_{l+u}$) and $M$. To verify the GPT ranking, we performed human evaluation on a

---

[9]The details are in Appendix A.3.1.

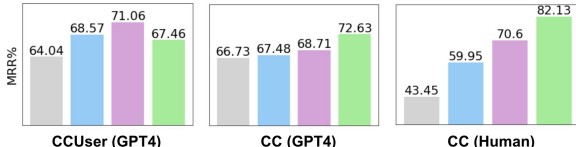

Figure 1: CCUser&CC GPT4 preference. We instructed GPT4 to give preference ranking for 4 AI-generated summaries (on 500 data points): M (not trained on CCUser), SALT$_l$, SALT$_{l+u}$, SALT$_{l+u}$+RSALT$_{l+u}$. (1) SALT$_{l+u}$ is most preferred by GPT4 on CCUser, (2) while SALT$_{l+u}$+RSALT$_{l+u}$ is most preferred by GPT4 on CC. (3) CC on human preference (on 25 data points) for M, SALT$_l$, SALT$_{l+u}$, and SALT$_{l+u}$+RSALT$_{l+u}$.

smaller set (25 data points). Human ranking agrees with the GPT4 ranking. In CCUser, GPT4 (on 500 data points) ranks SALT$_{l+u}$ higher than other variations, which is expected as SALT$_{l+u}$+RSALT$_{l+u}$ is also trained on the replay dataset. Because of privacy reasons, we did not do the human evaluation on CCUser. In Appendix Table 12, we show the prompt used with GPT4 for ranking the summaries. We show all the MRR scores for different models in our work in Figure 1.

## 7 Discussion: SALT vs RLHF

First, we argue that Human Edits is a more natural way to collect feedback from users as they fix AI-generated text for their workflow to improve generation. Collecting other forms of feedback that are not directly tied to the user's workflow will not scale as much, this is especially true in domains requiring expert domain knowledge and with nuanced user goals. Considering the cost, time, and availability of the experts, it is important to collect HF from the expert's daily workflow.

Second, we experiment with Direct Preference Optimization (DPO) (Rafailov et al., 2023) to compare the difference between RLHF and SALT while using a human edit feedback dataset. The training setup of DPO and SALT are similar, they are trained directly on the human preference dataset without training explicit reward models. We use $S_{AI}$ as the rejected summary and $S_E$ as the chosen summary and calculate the DPO loss $- L_{DPO}$, between them to train the model.

$$lRatio_\theta = log\, \pi_\theta(S_E|U) - log\, \pi_\theta(S_{AI}|U) \quad (9)$$

$$lRatio_{ref} = log\, \pi_{ref}(S_E|U) - log\, \pi_{ref}(S_{AI}|U) \quad (10)$$

$$L_{DPO} = -log\, \sigma(\beta * (lRatio_\theta - lRatio_{ref})) \quad (11)$$

where $\theta$ and $ref$ are the current and original model parameters. Table 7 shows the performance of DPO

| | Reward Acc | R1 | R2 | Rl | Meteor |
|---|---|---|---|---|---|
| SALT$_l$ | 0.368 | 0.381 | 0.203 | 0.371 | 0.292 |
| SALT$_{l+u}$ | 0.591 | 0.394 | 0.215 | 0.383 | 0.320 |
| DPO$_{beta=0.1}$ | 0.484 | 0.379 | 0.210 | 0.369 | 0.301 |
| DPO$_{beta=0.5}$ | 0.532 | 0.372 | 0.191 | 0.361 | 0.291 |

Table 7: SALT and DPO results on CCUser with GPT-2

for $\beta = \{0.1, 0.5\}$ on GPT-2[10] (117M parameters), with Rouge, Meteor, and Reward Accuracy (Reward Acc) on the CCUser test dataset. Reward Accuracy used in DPO[11] is the ratio of data points for which $chosen\ reward > rejected\ reward$.

$$chosen\ reward = \beta * (\pi_\theta(S_E|U) - \pi_{ref}(S_E|U)) \quad (12)$$

$$rejected\ reward = \beta * ((\pi_\theta(S_{AI}|U) - \pi_{ref}(S_{AI}|U)) \quad (13)$$

We find that DPO is better than $SALT_l$ which is just equivalent to likelihood training on $S_E$. This is expected since DPO also uses $S_{AI}$. However, DPO gets lower performance than $SALT_{l+u}$. When we change hyper-parameter $\beta$ to get higher Reward Accuracy, others (ROUGE, and Meteor) degrade, and vice versa. We think this is because, DPO penalizes the entire rejected summary, which is not suitable for human edit feedback, because most words in $S_{AI}$ and $S_E$ are the same. DPO does not explicitly consider such cases, and hence, it might be difficult for DPO to learn an implicit reward through $S_{AI}$ and $S_E$ without using the fine-grained relationship between their tokens. It is interesting to see that Reward Accuracy is higher for SALT than DPO, even though the SALT loss function does not explicitly maximize chosen and rejected log probability like DPO.

It should be noted that DPO was developed for using comparisons and not human edit feedback. For human edits feedback, a straightforward way to improve DPO could be to modify the loss function to use only the "negative tokens" in the rejected summary, which aligns with our SALT ideas.

## 8 Conclusion

In this work, we explore improving language models with Human Edits feedback, which can be collected scalably than others. Specifically, we propose the SALT training objective based on sequence alignment and unlikelihood training and show how to design Imitation Edits to reduce the need for expensive HF. We further show on human edits data, SALT performs better than a straightforward RLHF (DPO) approach.

---

[10]We used GPT because, at the time of this paper, DPO is only implemented on decoder-only models in Hugging Face

[11]https://huggingface.co/docs/trl/main/en/dpo_trainer

## 9 Limitations

In our experiments, we find that our method improves relatively smaller language models like T5. Due to the limitation of computational resources, we are not able to try our methods on larger language models. So we don't understand which HF (human feedback or human edit data) is better on LLMs. But like what we discussed in Section 1, Human-Edits have many unique advantages from an ML data point of view. Given that it's a natural way to collect feedback from users as they fix our AI-generated summaries for their workflow, many products in the industry can more easily use this HF approach and our SALT method to improve their text generation quality without too much extra effort. In addition, other HF methods should be explored more in various domains and models of various sizes so as to help the NLP community find the most suitable HF method in various scenarios.

Another point that has not been explored in this paper is LLM-in-the-loop. With the emergence of GPT3.5 and ChatGPT, LLM has shown a level close to or beyond human beings in many domains. In this paper, we did not use LLMs to conduct experiments similar to Human Edits (that is, treat the LLM as a human to modify $S_{AI}$ to get $S_{E(LLM)}$). Ideally, this would provide better Imitation-Edits to reduce HF costs. In addition to time and resource constraints, as we discussed in Section 1, data privacy issues make it hard for many practitioners in the industry to input their data into these third-party APIs or service websites for related experiments. LLM-in-the-loop is undoubtedly a worthwhile next step in the future, and we will study how to deal with related data privacy issues. This will also be a problem to be solved for many other tasks in medical and other privacy-oriented domains.

The current implementation of our methods also has some room for improvement. Our code currently only tries one global sequence alignment algorithm, the Needleman-Wunsch Algorithm. In fact, there are many alternatives that can help the model improve in different aspects. For example, how to improve factuality during LM's summaries is one key topic for both NLP and BioNLP community (Tang et al., 2022; Abacha et al., 2023b; Chang et al., 2023). Some previous work exploring language models and knowledge has shown that insufficient knowledge may lead to factual errors (Petroni et al., 2019; Sung et al., 2021; Yao et al., 2022a,b). So we can limit the scope of sequence alignment to the medical entities (Luo et al., 2022) or jargon (Kwon et al., 2022) to help the model focus more on important tokens during the training process to reduce hallucination further.

## 10 Ethics Statement

The methods related to unlikelihood training are very dependent on the quality of negative candidates. In this paper, we propose a very general framework to provide negative candidates, that is, to calculate the sequence alignment between $S_{AI}$ and Human-Edits or Imitation-Edits. There will be some potential problems in actual deployment: First of all, for Human-Edits, we don't know whether the user is modifying because of some kind of error in $S_{AI}$ or because of the user's personal preference. These two behaviors need to be distinguished in future research or actual deployment because the former data is more suitable for improving the problems of the model itself (such as some factual errors), and the latter data is more suitable for user-personalized training data. Secondly, whether for Human-Edits or Imitation-Edits, when a large number of complex Edits appear, the sequence alignment algorithm we currently use may not be able to get the correct negative candidates, resulting in rewards or penalties for wrong tokens. In the experiments in this paper, we use some filters to control the quality of the training data provided for unlikelihood training, but the reality will be very complicated. In addition to using similar filters in this paper, another solution is to directly track the users' changes as they edit the summary on the product, and the subsequent training steps will not change. But this will add a lot of extra overhead to the product engineering.

## Acknowledgements

We thank the Abridge AI for CC and CCUser data, as well as the professionals who performed the human evaluations. In addition, we also thank UMass BioNLP Lab for producing and providing us with a large batch of publicly available GPT Edits data for related work [12].

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

# A  Appendix

## A.1  SOAP Structure

The SOAP (Subjective, Objective, Assessment, and Plan) structure is commonly used by providers (Podder et al., 2021).

* The Chief Complaint section is a brief description of a patient's conditions and the reasons for the visit.

* The Subjective section is a detailed report of the patient's current conditions, such as source, onset, and duration of symptoms, mainly based on the patient's self-report. This section usually includes a history of present illness and symptoms, current medications, and allergies.

* The Objective section documents the results of physical exam findings, laboratory data, vital signs, and descriptions of imaging results.

* The Assessment section typically contains medical diagnoses and reasons that lead to medical diagnoses. The assessment is typically based on the content of the chief complaint and the subjective and objective sections.

* The Plan section addresses treatment plans based on the assessment.

## A.2  Implementation Details

Due to data privacy issues, we cannot disclose our CC and CCUser datasets. But for the reproduction of our methods, in the Appendix, we also use two general domain summarization datasets, CNN/Daily Mail (CNN) (See et al., 2017) and Extreme Summarization (XSum) (Narayan et al., 2018) to test the imitation-edit experiments.

The summarization model used in this paper is based on the publicly available T5-small model[13] and T5-large[14]. Note that the experimental results of our t5-large-based model are not real human edit feedback for the summaries it generates, because of some deployment and privacy issues, we can only collect the CCUser data (Human Edits) for t5-samll-based-model-generated summaries via our mobile app. Therefore, we put t5-large-related results only in the appendix. All the results in Section 6.1 are for our t5-small-based model. But overall, the patterns and findings are consistent on both t5-small and t5-large.

---

[13]https://huggingface.co/t5-small
[14]https://huggingface.co/t5-large

| Section | Subsection | Definition |
|---|---|---|
| Subjective | | |
| | Chief Complaint | Patient's primary motivation for the visit and type of visit |
| | Review of Systems | Patient's report of system-related health and symptoms |
| | Past Medical History | Patient's reported diagnoses/conditions (when and what, excluding laboratory and imaging results and surgeries) |
| | Past Surgical History | Patient's reported prior surgeries (what, when, where) |
| | Family Medical History | Conditions affecting patient's close genetic relatives |
| | Social History | Patient's alcohol, tobacco, and drug-related behaviors |
| | Medications | Patient's list of medications (not prescribed during visit) |
| | Allergies | Patient's list of allergies (primarily medicinal) |
| | Miscellaneous | Patient's clinically relevant social and other circumstances |
| Objective | | |
| | Immunizations | Vaccination record (not frequently discussed) |
| | Laboratory and Imaging Results | Clinician's discussion of laboratory/imaging results |
| Assessment | | |
| | Assessment | Synthesis of the reason for the visit and pertinent diagnosis |
| Plan | | |
| | Diagnostics & Appointments | Plan for future tests, appointments, or surgeries |
| | Prescriptions & Therapeutics | Plan for medications and therapeutics |

Table 8: Details of the SOAP structure used in our CC and CCUser datasets.

In this paper, we used '-1' for $w_{AI-C}$ [15] and 1 for $w_{AI-NC}$ in Equations 1, 2, 5. We trained $\text{M}_{CC}$ on the annotated Clinician Conversations (CC) dataset for 10000 steps and $\text{M}_{CNN}$ on CNN data with 100000 steps (batch size of 8). We then initialized the CCUser models $\text{SALT}_x$ with M and trained them on 70 human-edit notes for 1000 steps (batch size of 8)[16].

In Section 6.2.1 and A.3.2, we ran $M$ on both CC and CNN datasets' training data (See et al., 2017) to get the AI generated summaries $S_{AI}$, and we use the ground truth data as Imitation Edits on the seen data $S_I$. We then initialized $\text{SALT}_x$ from $\text{M}_{CC}$ and $\text{M}_{CNN}$ separately and trained on corresponding Imitation Edits with 1000 steps. We used CC-eval and CNN-eval to evaluate the models' performance.

In Section 6.2.2 and A.3.3, we sampled 3000 CC test data summaries (11812 data in total), 3000 CNN test data (11490 data in total), and 3000 XSum test data (11334 data in total) as Imitation Edits on the unseen data since we don't have the unseen training data in these datasets. Similarly, we initialized $\text{SALT}_x$ from M and trained on Imi-

tation Edits with 1000 steps. We took the remaining test data of CC-test, CNN-test, and XSum-test (Narayan et al., 2018) respectively as Imitation Edits evaluation data, and then used CC-eval and CNN-eval to evaluate the performance of the model in the original data.

In all our evaluations, we used a beam size of 4, no-repeat-ngram-size=2, and minimum length and maximum length of sentences were set as (10, 100). We used five different random seeds to sample training data for all our experiments, and the scores reported in the tables are the average of these random seeds.

### A.3 Imitation Edits Experiments

#### A.3.1 Imitation Edits smoothing function

Although both come from humans, $S_E$ and $S_I$ are fundamentally different in their relationship with $S_{AI}$. The former is modified from $S_{AI}$ while humans generate the latter from scratch. Therefore, $S_E$ is directly dependent on $S_{AI}$, but $S_I$ is not. Consequently, even though $S_E$ and $S_I$ are dependent on the same data as input, the differences between $S_{AI}$ and $S_I$ are likely to be larger than between $S_{AI}$ and $S_E$. We can see this difference in the average percentage of changed tokens $- \mathbb{1}_{E-C}$ and $\mathbb{1}_{I-C}$ is 1, the former (6.17%) is much lower than the latter (45.59%). Hence, after we do se-

---

[15]For $w_{AI-C}$, we used -1.2 for $\text{SALT}_{l_i}$ and -0.5 for $\text{SALT}_{l_d}$, and all other $\text{SALT}_x$ and $\text{RSALT}_x$ use -1.

[16]We did all the experiments with 1 NVIDIA Tesla P100 GPU - 16 GB memory, with Adam optimizer – betas=(0.9,0.999), epsilon=1e-08, learning rate=5e-05.

| | CCUser$_{eval}$ | | CC$_{eval}$ | |
|---|---|---|---|---|
| | R1 | U-f | R1 | U-f |
| SALT$_l$ | 57.77(±0.28) | 61.02(±1.06) | 34.27(±0.21) | 46.45(±0.52) |
| SALT$_{l_d}$ | 57.70(±0.25) | 61.06(±0.86) | 34.46(±0.31) | 46.58(±0.33) |
| SALT$_{l_i}$ | 57.84(±0.36) | 60.81(±0.79) | 34.68(±0.25) | 46.77(±0.71) |
| SALT$_u$ | 57.57(±0.66) | 61.09(±1.33) | 34.47(±0.44) | 46.64(±0.51) |
| SALT$_{l+u}$ | 58.39(±0.57) | 62.13(±1.03) | 34.79(±0.30) | 47.06(±0.47) |
| SALT$_l$+RSALT$_l$ | 59.57(±0.47) | 62.52(±0.98) | 35.55(±0.32) | 48.25(±0.60) |
| SALT$_{l+u}$+RSALT$_l$ | 59.60(±0.52) | 62.57(±1.34) | 35.43(±0.23) | 48.20(±0.68) |
| SALT$_l$+RSALT$_{l+u}$ | 59.88(±0.43) | 62.60(±0.85) | 36.24(±0.36) | 48.42(±0.41) |
| SALT$_{l+u}$+RSALT$_{l+u}$ | 60.43(±0.61) | 63.44(±0.92) | 36.26(±0.40) | 48.69(±0.59) |

Table 9: 95% Confidence interval results for Table 3

| | CCUser$_{eval}$ | | CC$_{eval}$ | |
|---|---|---|---|---|
| | R1 | U-f | R1 | U-f |
| $M$ | 40.48 | 42.22 | 37.21 | 47.12 |
| SALT$_l$ | 64.28 | 64.90 | 36.67 | 48.51 |
| SALT$_{l+u}$ | 63.47 | 64.95 | 37.10 | 48.66 |
| SALT$_l$+RSALT$_l$ | 62.49 | 62.86 | 37.58 | 49.87 |
| SALT$_{l+u}$+RSALT$_{l+u}$ | 64.71 | 64.53 | 37.87 | 50.02 |

Table 10: T5-large results on CCUser dataset. T5-large is also first fine-tuned on the CC dataset and then fine-tuned on the CCUser dataset. Note that CCUser data is collected only for T5-small, so it's not a real Human Edits dataset for T5-large.

quence alignment between $S_I$ and $S_{AI}$, we perform a two-step post-processing operation to ensure the training stability, First, we only penalize consecutive tokens ($> 1$) in $S_{AI}$ that are not aligned with $S_I$, for eg., the $\mathbb{1}_{AI-NC}(t) = [1, 0, 1, 1, 0, 0, 1]$, becomes $[1, 1, 1, 1, 0, 0, 1]$, and the corresponding change is made to $\mathbb{1}_{AI-C}(t)$. This smoothing is to reduce the impact of less important negative tokens, for e.g., punctuation and word plural, as they are more frequently present in such single negative tokens. On the contrary, consecutive negative tokens are more likely to represent important errors (e.g., hallucination and missing information). Second, we discard data with more than 60% of the tokens being 0 in the indicator function $\mathbb{1}_{AI-NC}$, which helps us to reduce the percentage of changed tokens from 45.59% to 19.07% with an acceptable amount of data lost (21.38%).

**A.3.2 Imitation Edits using seen data**

We use the training data from two datasets, CC and CNN, to experiment with the effects of SALT and Imitation Edits on seen data. First, for the CC dataset, the results in Table 5 show that continu-ing to use likelihood loss on the training dataset to train the already convergent $M$ does not improve the performance and leads to overfitting. However, when we use $S_I$ as imitation-edit data and do SALT training on it with $S_{AI}$, we can see an improvement. Second, we see similar results for the CNN dataset. Even though there is no performance degradation arising from overfitting for $SALT_l$, doing SALT training with $S_I$ and $S_{AI}$ can improve the performance more than using just the likelihood training. These results show that we can get additional improvement on the model by continuing to train it with SALT on the seen dataset even if the model is already converged (on the seen/original training data). Third, different from previous human-edit results, $SALT_u$ of CC is better than $SALT_{l+u}$. We think this is because $M$ has started to overfit on CC data, so continuing to add likelihood to the original training data reduces the scores.

**A.3.3 Imitation Edits using unseen data**

We use a part of the test dataset (not used in evaluation) from CC, CNN and XSum to experiment with the effects of SALT and Imitation Edits on unseen data. In Table 6, we show three experimental results. In the first experiment, we take $M$ (trained on CC-train) and train it with a part of the CC-test as the imitation-edit data with SALT. In the second experiment, we take $M$ (trained on CNN-train) and train it with a part of the CNN-test as imitation-edit data. In the third experiment, we take $M$ (trained on CNN-train) and train it with a part of the XSum-test as imitation-edit data. In the three experiments, we took the remaining test data of the CC-test, CNN-test, and XSum-test, respectively, to evaluate the model performance in new imitation-edit data and then used CC-eval and CNN-eval to evaluate the model performance in

| | (CC-test) <CC-train> | | | | (CNN-test) <CNN-train> | | | | (XSum-test) <CNN-train> | | | |
|---|---|---|---|---|---|---|---|---|---|---|---|---|
| | $CC_{test-r}$ | | $CC_{eval}$ | | $CNN_{test-r}$ | | $CNN_{eval}$ | | $XSum_{test-r}$ | | $CNN_{eval}$ | |
| | R1 | U-f | R1 | U-f | R1 | R2 | R1 | R2 | R1 | R2 | R1 | R2 |
| M | 36.01 | 58.15 | 36.07 | 48.97 | 36.44 | 15.28 | 36.99 | 15.49 | 17.40 | 2.36 | 36.24 | 15.35 |
| $SALT_l$ | 36.09 | 57.55 | 36.14 | 48.50 | 36.97 | 15.39 | 37.29 | 15.29 | 26.56 | 7.22 | 26.02 | 8.79 |
| $SALT_{l+u}$ | 36.57 | 58.12 | 36.28 | 48.84 | 37.59 | 16.04 | 38.17 | 16.35 | 27.03 | 7.27 | 28.64 | 9.27 |
| $SALT_l$+$RSALT_{l+u}$ | 36.73 | 57.48 | 36.61 | 48.61 | 37.57 | 16.02 | 38.39 | 16.65 | 25.56 | 7.07 | 36.97 | 15.57 |
| $SALT_{l+u}$+$RSALT_{l+u}$ | 36.74 | 58.48 | 36.65 | 48.77 | 37.73 | 16.10 | 38.42 | 16.65 | 26.10 | 6.94 | 37.71 | 16.02 |

Table 11: Imitation Edits experiments: Here the imitation-edit data comes from a subset of the corresponding test dataset (we don't use them in the table for metrics) which $M$ has never seen before. () and <> show the training data used by SALT and RSALT respectively.

the original data.

In imitation-edit evaluation results ($CC_{test-r}$, $CNN_{test-r}$, $XSum_{test-r}$) of Table 6, $SALT_{l+u}$ has better performance than the baseline method $SALT_l$ in all three experiments, which is consistent with our results using human-edit data in Table 3. In the original data evaluation results ($CC_{eval}$, $CNN_{eval}$) of Table 6, although there was no forgetting problem in the first two experiments, $SALT_{l+u}$ still has a higher score than the baseline model $SALT_l$. In the third experiment, we successfully imitated the forgetting problem similar to CC and CCUser by using the distribution difference between CNN and XSum. Similar to the results in Table 3, $SALT_{l+u}$ can alleviate the forgetting problem to a certain extent while improving the performance on the new dataset.

## A.4 More Discussion

**Why does SALT work?** First, SALT makes good use of the $S_{AI}$ data. From the perspective of data augmentation, $S_E$ provides a new ground truth summary from the user, and the users also verify the remaining tokens in $S_{AI}$. SALT helps the model to use all the tokens in both $S_E$ and $S_{AI}$, which greatly improves the utilization of human-edit data. Second, SALT gives the model more objectives. Using $S_{AI}$ in SALT makes the model not just "be close to the correct distribution" as in the likelihood training, but also "be far away from a negative distribution". Thus, we can teach the model to avoid making the same mistakes again, which has a special meaning for the user (Assumption 1).

**Human Edits and Imitation Edits** Even though SALT can be used with human-edit data or imitation-edit data to improve the summarization models, our experiments are not enough to conclude that Imitation Edits can completely replace

In this task, we ask for your expertise in annotating the quality of system-generated SOAP notes by machine learning models. Mainly we provide a conversation snippet and a human-written reference SOAP note for the respective snippet, along with system-generated summaries, and ask for your preference.

Output your ranking for system-generated summaries. Use the following format, and do not add any other text.

Some examples:
a > b > c > d
d > c > b > a

Conversation snippet:
[*conversation*]

Human written reference SOAP note for the respective snippet:
[*reference*]

System-generated summaries:
1. [*summary1*]
2. [*summary2*]
3. [*summary3*]
4. [*summary4*]

Now, output your ranking:

Table 12: GPT4 Prompt for preference ranking.

Human Edits. Using Imitation Edits is essentially a kind of data augmentation method during training. But, when we have edits to our model's original output from our real users, we have the unique opportunity to improve model output according to their individual expectations. SALT can model such information during the training and help the model have more appropriate behaviors to serve the users better in a more data-efficient way.

**SALT and RLHF** We discuss SALT and DPO in the Section 7. Regarding the relationship between SALT and other RLHFs, we have some preliminary discussions here, and they need follow-up work to demonstrate. It seems that SALT keeps most of the advantages and disadvantages of DPO against

PPO. Often, no reinforcement learning means more stable and easy training (and hyper-tuning). Our human eval also shows that SALT can make models more aligned with human preference without explicit reward models, which is the same with DPO. Also, it's questionable whether a good explicit reward model can be learned from $S_{AI}$ and $S_E$ since it's not as easy as positive or negative movie reviews to distinguish. For limitation, How does the SALT model generalize out of distribution, compared with PPO with an explicit reward function? For example, standard RLHF methods can leverage additional unlabeled prompts by labeling LM generations with the learned reward model. Can training with self-labeling from the SALT similarly make effective use of unlabeled prompts? Other papers like RAFT and RRHF use explicit reward models to filter high-score data points for SFT. Whether we can train a good reward model as a good filter is also a big question here. Another difference is that we will make full use of all data points ($S_{AI} + S_E$) during the training, but they will only use high-quality ones ($S_E$) and discard the rest ($S_{AI}$). So theoretically, we use data more efficiently and model more information from $S_{AI}$.