# OpenReview forum: "Improving Summarization with Human Edits"
_EMNLP/2023/Conference — EMNLP 2023 Main_

### Official Review · Reviewer_sCPx · 2023-08-01

**Typos Grammar Style And Presentation Improvements:** 1) 04
**Soundness:** 3

**Excitement:**

4: Strong: This paper deepens the understanding of some phenomenon or lowers the barriers to an existing research direction.

**Paper Topic And Main Contributions:**

The paper presents an approach (referred to as SALT) to incorporate human edits for improving summarization of clinical texts. The approach relies on a combination of likelihood and unlikelihood training to reduce the probability of tokens that are edited and increase the probability of tokens that are retained and that of the new tokens that are added by the human user. The authors perform experiments using two datasets and show improvements in summarization across a variety of metrics.

Overall, I believe that the idea is interesting and naturally suited for the problem. However, I have concerns about some parts of the approach and the presentation.

**Questions For The Authors:**

1) Is there a thought about doing this process repeatedly? Getting human feedback, retraining the model, and then getting further feedback?
Can this iterative process help to reduce costs by being more data efficient?

2) Section 3: This presentation can be drastically improved. a) Lacks citations - unclear where the datasets are coming from, b) Not sure what the AI is in Section 3.2, c) The naming of CC and CCUser can be improved, d) Section 3 needs some introduction that connects the datasets to how they are being used.

3) Section 4: a) A figure might help. b) What exactly is the point of E-C and E-NC? This does not correspond to the 3 points described in 269-275. Why would one want to reduce the prob. of E-C tokens (as per Eq 2)? What am I missing here?

4) Line 491: Was any statistical test performed? Otherwise, the usage of the word might be misleading.



**Reasons To Accept:**

1) Well-motivated. I am convinced that human edits are a better form of incorporating human feedback than ratings. It especially holds value since the data can be naturally obtained from the user workflows.

2) Using unlikelihood training to incorporate the human edits (269-275) makes sense to me and is sound - the presentation can be improved though (see below).

3) The experiments are comprehensive, with sufficient analysis and ablations.

**Reasons To Reject:**

1) The presentation of Sections 3 and 4 is currently poor. I have questions about the datasets and the approach itself (see questions / suggestions below).

2) The imitation edits part is not convincing to me - given the diversity in generation, what is the point of aligning the AI-generated output to the ground truth? How to ensure that the AI-generated tokens that are not in ground truth are actually bad? The current experiments should include some statistics around this alignment between AI-generated summaries and the ground truth. Maybe putting some constraints, like only considering entities and their missing attributes in the AI generated summary, etc. might help.

**Reproducibility:**

4: Could mostly reproduce the results, but there may be some variation because of sample variance or minor variations in their interpretation of the protocol or method.

**Reviewer Confidence:**

4: Quite sure. I tried to check the important points carefully. It's unlikely, though conceivable, that I missed something that should affect my ratings.

---

> ### Author Rebuttal · Authors · 2023-08-29
>
> ### Reasons To Reject:
>
> **Q1:** The presentation of Sections 3 and 4 is currentlnotesor. I have questions about the datasets and the approach itself (see questions / suggestions below).
>
> **Answer:** Thanks to the reviewer's point here. We will revise these two sections in our final version based on your suggestions.
>
>
> **Q2:** The imitation edits part is not convincing to me - given the diversity in generation, what is the point of aligning the AI-generated output to the ground truth? How to ensure that the AI-generated tokens that are not in ground truth are actually bad? The current experiments should include some statistics around this alignment between AI-generated summaries and the ground truth. Mhuman editsg some constraints, like only considering entities and their missing attributes in the AI-generated summary, etc. might help.
>
>
> **Answer:** Thanks to the reviewer for this suggestion. In lines 570 to 587 of the paper, we discussed how to use a simple filter to control the quality of such imitation edits. Lines 707-715 of the limitation section already refer to your suggestion of "only considering entities and their missing attributes in the AI-generated summary".
>
> In general, we agree that the current imitation edits have significant shortcomings, in addition to the above two points, we found that GPT4 can provide "human-like edits" data very well, so using GPT4 edits as imitation edits is another option. Here we provide the experimental results based on GPT4 edits on the public dataset (also for reproduction). Here are the details:
>
> We couldn't find a big enough public dataset with (doctor-patient conversation, note) and human edits information to train a summarization model, so we experimented on the MIMIC-III after-visit-summary generation dataset [1] [2] for all SALT experiments (We will release our SALT code, scores, and output results of different models on this public dataset). In addition, we use GPT4 to edit the model output to get "human edits". The dataset has 31,895 (Clinical note, AVS) pairs for this summarization task. We first fine-tune the GPT2 on this dataset, and then ask GPT4 to edit the trained-GPT2 generated summaries based on the input clinical notes and reference AVS summaries. GPT4 can easily provide such “human-edits” without too much prompt engineering. So we have (Clinical note, GPT2 generated summary, GPT4-edit summary) for SALT. Here are the results:
>
> |**Results with GPT2**|**Rewards ACC**|**Rouge1**|**Rouge2**|**Rougel**|**Meteor**
> |-------|-------|-------|-------|-------|-------|
> |**SALT(l)**|**0.596**|**0.500**|**0.366**|**0.487**|**0.455**|
> |**SALT(l_d)**|**0.586**|**0.512**|**0.378**|**0.498**|**0.471**|
> |**SALT(l_i)**|**0.677**|**0.502**|**0.369**|**0.489**|**0.457**|
> |**SALT(u)**|**0.860**|**0.462**|**0.317**|**0.447**|**0.406**|
> |**SALT(l+u)**|**0.777**|**0.511**|**0.375**|**0.497**|**0.468**|
>
> The results here are in line with paper results.
>
> [1] Generation of Patient After-Visit Summaries to Support Physicians
> https://aclanthology.org/2022.coling-1.544/
>
> [2] Learning to Revise References for Faithful Summarization
> https://aclanthology.org/2022.findings-emnlp.296.pdf
>
>
>
>
> ### Questions For The Authors:
> **Question 1** Is there a thought about doing this process repeatedly? Getting human feedback, retraining the model, and then getting further feedback? Can this iterative process help to reduce costs by being more data efficient?
>
> **Answer:** Thanks to the reviewer for this suggestion, this is a very interesting topic and a project we are currently working on. Specifically, we used the human eval data collected from the product before, so it was impossible to collect human edits for the new coming batch to update the model parameters using SALT in real-time, and then collect human edits for the next batch (continuously loop until the human is satisfied). This is an experiment we want to conduct in the future, but it may require a lot of manpower to really participate in the human-in-the-training loop.
>
> The high-quality edits of GPT4 above allow us to use GPT4 to simulate the above process and achieve GPT4-in-the-training-loop (GPT4 can be regarded as a teacher model here). This is our ongoing experiment to support the evidence that we let experts enter the training loop in the future, and the relevant results will be shown in the next paper.
>
>
>
> **Question 2** Line 491: Was any statistical test performed? Otherwise, the usage of the word might be misleading.
>
> **Answer:** Thanks to the reviewer's point here. We add 95% Confidence interval results here:
>
> |**Methods**|**CCUser_R1**|**CCUser_Uf**|**CC_R1**|**CC_Uf**|
> |-------|-------|-------|-------|-------|
> |**SALT(l)**|**57.77** (±0.28)|**61.02** (±1.06)|**34.27** (±0.21)|**46.45** (±0.52)|
> |**SALT(l_d)**|**57.70**  (±0.25) |**61.06** (±0.86)|**34.46**  (±0.31)|**46.58** (±0.33)|
> |**SALT(l_i)**|**57.84**  (±0.36)|**60.81** (±0.79)|**34.68** (±0.25)|**46.77** (±0.71)|
> |**SALT(u)**|**57.57** (±0.66)|**61.09** (±1.33)|**34.47** (±0.44)|**46.64**(±0.51)|
> |**SALT(l+u)**|**58.39** (±0.57)|**62.13** (±1.03)|**34.79** (±0.30)|**47.06** (±0.47)|
> |**SALT(l) +RSALT(l)**|**59.57** (±0.47)|**62.52** (±0.98)|**35.55** (±0.32)|**48.25** (±0.60)|
> |**SALT(l+u) +RSALT(l)**|**59.60** (±0.52)|**62.57** (±1.34)|**35.43** (±0.23)|**48.20** (±0.68)|
> |**SALT(l) +RSALT(l+u)**|**59.88** (±0.43)|**62.60** (±0.85)|**36.24** (±0.36)|**48.42** (±0.41)|
> |**SALT(l+u) +RSALT(l+u)**|**60.43** (±0.61)|**63.44** (±0.92)|**36.26** (±0.40)|**48.69** (±0.59)|

---

### Official Review · Reviewer_88MQ · 2023-08-04

**Soundness:** 3

**Excitement:**

4: Strong: This paper deepens the understanding of some phenomenon or lowers the barriers to an existing research direction.

**Paper Topic And Main Contributions:**

The paper proposes a method to finetune a summarization model leveraging human edits (or imitation edits) that can steer an initial system summary towards a better reference. The method, named SALT, relies on improving the likelihood of tokens inserted in the edit, while lowering the likelihood of tokens removed in the edit. Further, to avoid catastrophic forgetting, the model is jointly finetuned on the original dataset (replay-based SALT, or RSALT). The model presents results based on ROUGE, UMLS-F1 (a medical-domain specific score), and SAGE (to focus on the removal of undesired tokens). These experiments are conducted on both the original CC dataset as well as a newly introduced (and smaller) CCUser. Experiments confirm that RSALT results leads to metric improvements.

**Questions For The Authors:**

Your work is great, yet proposing your own models, on your own dataset, with your own metrics, could be seen as siloed.
Connecting your work to prior work is an important step towards giving the reader of your work a nuanced understanding of how your work fits within the research space. Do you know of (external) baselines you could include which would contextualize your work?

**Reasons To Accept:**

- The paper addresses a novel task within summarization: how to effectively integrate user feedback -- in the form edits -- to improve system output summaries.
- The method is complex and seem to yield gains on several metrics. In particular, the experiments reveal that with RSALT, the model is able to maintain (or close) performance on the original dataset, while yielding improvements on the new task.

**Reasons To Reject:**

- The experiment section lacks real external comparisons. Currently the only systems that are compared are the full system, and several ablated versions of the method. In order to understand the method, it would be important to compare to prior work. Prior work in edit-based model could interesting to compare (the following tutorial from NAACL 2022 might be a good place to find pointers: https://text-editing.github.io/ ). In essence, a reader needs to understand: is this method an efficient way to integrate human feedback/edits compared to other methods, not just which components of the proposed system work/do not work (ablation study).
- Some reorganization of the content, and improving priorities within the main paper might be beneficial. For instance, the authors mention a human evaluation, yet there is no space in the paper for its results. When going into the Appendix, one finds out that such results are positive and interesting, even though they are themselves buried in a Table with GPT4-based experiments. Why not give the human evaluation results importance? Isn't this the gold standard?
- In general with the method/evaluation, it seems like the authors are reinventing the wheel a bit. The method is quite complex, with many variants (when in the end, the variant with all components seem to win it all), the SAFE metric proposed seems a bit ad-hoc. Why not use well-known, existing edit-based metrics like SARI (or newer variants) from edit-based tasks (such as simplification), they fulfill similar objectives (measuring distance from an bad reference, and similarity to a good reference).

**Reproducibility:**

2: Would be hard pressed to reproduce the results. The contribution depends on data that are simply not available outside the author's institution or consortium; not enough details are provided.

**Reviewer Confidence:**

4: Quite sure. I tried to check the important points carefully. It's unlikely, though conceivable, that I missed something that should affect my ratings.

---

> ### Author Rebuttal · Authors · 2023-08-29
>
> ### Reasons To Reject:
>
> **Q1:** The experiment section lacks real external comparisons. Currently the only systems that are compared are the full system, and several ablated versions of the method. In order to understand the method, it would be important to compare to prior work. Prior work in edit-based model could interesting to compare (the following tutorial from NAACL 2022 might be a good place to find pointers: https://text-editing.github.io/ ). In essence, a reader needs to understand: is this method an efficient way to integrate human feedback/edits compared to other methods, not just which components of the proposed system work/do not work (ablation study).
>
> **Answer:** Thanks to the reviewer's point here. In order to solve this question, we experimented on the most representative work of RLHF recently, DPO (Direct Preference Optimization), on our CCUser dataset, because the settings of the two are similar, that is, training directly on the human preference dataset , without training explicit reward model. Specifically, we use S_AI as the rejected summary, S_E as the chosen summary and calculate the DPO loss to train the model.
>
> |**Results with GPT2**|**Rewards ACC**|**Rouge1**|**Rouge2**|**Rougel**|**Meteor**
> |-------|-------|-------|-------|-------|-------|
> |**SALT(l)**|**0.368**|**0.381**|**0.203**|**0.371**|**0.292**|
> |**SALT(l_d)**|**0.430**|**0.388**|**0.210**|**0.376**|**0.302**|
> |**SALT(l_i)**|**0.390**|**0.389**|**0.214**|**0.376**|**0.320**|
> |**SALT(u)**|**0.522**|**0.391**|**0.216**|**0.380**|**0.310**|
> |**SALT(l+u)**|**0.591**|**0.394**|**0.215**|**0.383**|**0.320**|
> |**DPO(beta=0.1)**|**0.484**|**0.379**|**0.210**|**0.369**|**0.301**|
> |**DPO(beta=0.5)**|**0.532**|**0.372**|**0.191**|**0.361**|**0.291**|
>
> Here are our findings:
>
> - It is difficult for DPO to learn an implicit reward through S_AI and S_E because most words in S_AI and S_E are the same. So we find the reward metrics (reward ACC or margin) and other metrics (ROUGE, Factuality score) become negatively correlated. If we change hyper-parameters to get high reward metrics, the ROUGE and Factuality score becomes extremely bad, and its final generated summary becomes a mess. On the contrary, if we want to keep a ROUGE, Factuality score, the reward metrics will not improve at all (similar to SFT training). It’s easy to explain from the DPO loss function:
>
> 	> pi_logratios = policy_chosen_logps - policy_rejected_logps
>
> 	> ref_logratios = reference_chosen_logps - reference_rejected_logps
>
> 	> logits = pi_logratios - ref_logratios
>
> 	> DPO loss = --F.logsigmoid(self.beta * logits)
>
> So DPO considered penalizing the whole rejected summary, which is not a suitable behavior with human edit feedback. Again, most words in S_AI and S_E are the same, so DPO does not consider such cases.
>
> - SALT can be treated as an extension or variance of DPO, especially for human edit feedback. One straightforward solution to improve DPO on human edit feedback is to target “negative tokens'' in the rejected summary and calculate DPO loss, which aligns with our SALT ideas. After doing sequence alignment to find negative tokens, we use likelihood and unlikelihood loss for different tokens, but DPO loss can also be a good choice. We also calculate DPO rewards when we train the model with SALT, and interestingly, we find that, even though SALT loss function doesn’t explicitly maximize chosen and rejected logps, we still see similar improvement for DPO rewards metrics (e.g., ACC or margin).
>
> 	> chosen_rewards = self.beta * (policy_chosen_logps - reference_chosen_logps)
>
> 	> rejected_rewards = self.beta * (policy_rejected_logps - reference_rejected_logps)
>
> We will update these small experiments in the appendix and discuss our findings. We also treat this as our future work with more deep and comprehensive experiments.
>
> - We don’t have enough time to try all recent RLHF methods, but here is some thoughts:
>
> 	> It seems that SALT keeps most of the advantages and disadvantages of DPO against PPO. Often, no reinforcement learning means more stable and easy training (and hyper-tuning). Our human eval also shows that SALT can make models more aligned with human preference without explicit reward models, which is the same with DPO. Also, it’s questionable whether a good explicit reward model can be learned from S_AI and S_E since it’s not as easy as positive/negative movie reviews to distinguish. For limitation, How does the SALT model generalize out of distribution, compared with PPO with an explicit reward function? For example, standard RLHF methods can leverage additional unlabeled prompts by labeling LM generations with the learned reward model. Can training with self-labeling from the SALT similarly make effective use of unlabeled prompts?
>
> 	> Other papers like RAFT and RRHF use explicit reward models to filter high-score data points for SFT. Whether we can train a good reward model as a good filter is also a big question here. Another difference is that, we will make full use of all data points (S_AI + S_E) during the training, but they will only use high quality one (S_E) and discard the rest (S_AI). So theoretically we use data more efficiently and model more information from S_AI.
>
> It’s amazing to see the RLHF community grow up so quickly! We will explore and try more experiments to compare SALT with all these new methods (DPO/RAFT/RRHF) in the future.
>
>
> **Q2:** Some reorganization of the content, and improving priorities within the main paper might be beneficial. For instance, the authors mention a human evaluation, yet there is no space in the paper for its results. When going into the Appendix, one finds out that such results are positive and interesting, even though they are themselves buried in a Table with GPT4-based experiments. Why not give the human evaluation results importance? Isn't this the gold standard?
>
>
> **Answer:** Thanks to the reviewer for this suggestion. Yes, human eval is most important in generation task. Unfortunately, since CCUser is the product data, we can't do human eval on it due to the privacy restrictions, that's why we only do human eval with CC (human annotated data). But we did GPT4 eval on both CC and CCUser, and the results were as expected. At the same time, due to data privacy restrictions, we cannot provide specific case studies of different behavior after SALT training on CCUser in the paper. These reasons made us decide to just put them in the appendix when submitting.
>
>
> **Q3:** In general with the method/evaluation, it seems like the authors are reinventing the wheel a bit. The method is quite complex, with many variants (when in the end, the variant with all components seem to win it all), the SAFE metric proposed seems a bit ad-hoc. Why not use well-known, existing edit-based metrics like SARI (or newer variants) from edit-based tasks (such as simplification), they fulfill similar objectives (measuring distance from an bad reference, and similarity to a good reference).
>
> **Answer:** Thanks to the reviewer for this suggestion. Actually, we find DPO defined reward ACC is also a good and simple metric (try to see whether chosen_rewards > rejected_rewards) for a similar goal, and we report above, which also supports our previous results. We will explore and try more related metrics for this goal.
>
>
>
>
> ### Questions For The Authors:
> **Question** Your work is great, yet proposing your own models, on your own dataset, with your own metrics, could be seen as siloed. Connecting your work to prior work is an important step towards giving the reader of your work a nuanced understanding of how your work fits within the research space. Do you know of (external) baselines you could include which would contextualize your work?
>
>
>
> **Answer:** The answer here is the same with point1 in Reasons To Reject.

---

### Official Review · Reviewer_ZppW · 2023-08-10

**Typos Grammar Style And Presentation Improvements:** 1. 256-257
**Soundness:** 3

**Excitement:**

3: Ambivalent: It has merits (e.g., it reports state-of-the-art results, the idea is nice), but there are key weaknesses (e.g., it describes incremental work), and it can significantly benefit from another round of revision. However, I won't object to accepting it if my co-reviewers champion it.

**Missing References:**

Between 075 and 098, the paradigm of learning from human edits is explained. I feel that some related works could be mentioned here, in the domain of summarization, or perhaps other NLP tasks.

**Paper Topic And Main Contributions:**

This work proposes to apply unlikelihood training to human-edited summary data in the clinical domain. The author in addition proposed to experiment with gold reference as an imitation edited data. There are a good amount of experiments and evaluations, but the improvement is not consistent or significant.

**Questions For The Authors:**

I find the data split not very clear. In 243-246, "we randomly select 70 notes from 7 physicians as training dataset, 10 for each physician, and divide the remaining 145 notes into 245 evaluation and test sets." Why is it important to mention the number of physicians covered in the training data? I also wonder if the training and test sets overlap in terms of being produced by the same physician. If so there might be implicit bias such as the use of words or the edit style.

**Reasons To Accept:**

1. The idea of (un)likelihood training using the original output as well as human edits is reasonable and could inspire other NLP applications.
2. The formation of gold reference as imitation edited data is interesting, although it might be overkill.

**Reasons To Reject:**

1. Due to the way the data is processed, which uses humans, Google APIs, and the in-house model, it is hard to reproduce the work.
2. Empirically, the improvements observed are marginal: for a baseline ROUGE of ~50, the improvement is ~0.5 (1%). The result possibly does not strongly encourage the adoption of the proposed method.
3. Further, the paper does not have a sufficient comparison with related works through experiments either.

 (Update: I think 2 and 3 are partially addressed as the authors added more experiments to compare with existing literature, and reported significance testing of existing results in the rebuttal.)

**Reproducibility:**

2: Would be hard pressed to reproduce the results. The contribution depends on data that are simply not available outside the author's institution or consortium; not enough details are provided.

**Reviewer Confidence:**

2: Willing to defend my evaluation, but it is fairly likely that I missed some details, didn't understand some central points, or can't be sure about the novelty of the work.

---

> ### Author Rebuttal · Authors · 2023-08-29
>
> ### Reasons To Reject:
>
> **Q1:** Further, the paper does not have a sufficient comparison with related works through experiments either.
>
> **Answer:** Thanks to the reviewer's point here. In order to solve this question, we experimented on the most representative work of RLHF recently, DPO (Direct Preference Optimization), on our CCUser dataset, because the settings of the two are similar, that is, training directly on the human preference dataset , without training explicit reward model. Specifically, we use S_AI as the rejected summary, S_E as the chosen summary and calculate the DPO loss to train the model.
>
> |**Results with GPT2**|**Rewards ACC**|**Rouge1**|**Rouge2**|**Rougel**|**Meteor**
> |-------|-------|-------|-------|-------|-------|
> |**SALT(l)**|**0.368**|**0.381**|**0.203**|**0.371**|**0.292**|
> |**SALT(l_d)**|**0.430**|**0.388**|**0.210**|**0.376**|**0.302**|
> |**SALT(l_i)**|**0.390**|**0.389**|**0.214**|**0.376**|**0.320**|
> |**SALT(u)**|**0.522**|**0.391**|**0.216**|**0.380**|**0.310**|
> |**SALT(l+u)**|**0.591**|**0.394**|**0.215**|**0.383**|**0.320**|
> |**DPO(beta=0.1)**|**0.484**|**0.379**|**0.210**|**0.369**|**0.301**|
> |**DPO(beta=0.5)**|**0.532**|**0.372**|**0.191**|**0.361**|**0.291**|
>
> Here are our findings:
>
> - It is difficult for DPO to learn an implicit reward through S_AI and S_E because most words in S_AI and S_E are the same. So we find the reward metrics (reward ACC or margin) and other metrics (ROUGE, Factuality score) become negatively correlated. If we change hyper-parameters to get high reward metrics, the ROUGE and Factuality score becomes extremely bad, and its final generated summary becomes a mess. On the contrary, if we want to keep a ROUGE, Factuality score, the reward metrics will not improve at all (similar to SFT training). It’s easy to explain from the DPO loss function:
>
> 	> pi_logratios = policy_chosen_logps - policy_rejected_logps
>
> 	> ref_logratios = reference_chosen_logps - reference_rejected_logps
>
> 	> logits = pi_logratios - ref_logratios
>
> 	> DPO loss = --F.logsigmoid(self.beta * logits)
>
> So DPO considered penalizing the whole rejected summary, which is not a suitable behavior with human edit feedback. Again, most words in S_AI and S_E are the same, so DPO does not consider such cases.
>
> - SALT can be treated as an extension or variance of DPO, especially for human edit feedback. One straightforward solution to improve DPO on human edit feedback is to target “negative tokens'' in the rejected summary and calculate DPO loss, which aligns with our SALT ideas. After doing sequence alignment to find negative tokens, we use likelihood and unlikelihood loss for different tokens, but DPO loss can also be a good choice. We also calculate DPO rewards when we train the model with SALT, and interestingly, we find that, even though SALT loss function doesn’t explicitly maximize chosen and rejected logps, we still see similar improvement for DPO rewards metrics (e.g., ACC or margin).
>
> 	> chosen_rewards = self.beta * (policy_chosen_logps - reference_chosen_logps)
>
> 	> rejected_rewards = self.beta * (policy_rejected_logps - reference_rejected_logps)
>
> We will update these small experiments in the appendix and discuss our findings. We also treat this as our future work with more deep and comprehensive experiments.
>
> - We don’t have enough time to try all recent RLHF methods, but here is some thoughts:
>
> 	> It seems that SALT keeps most of the advantages and disadvantages of DPO against PPO. Often, no reinforcement learning means more stable and easy training (and hyper-tuning). Our human eval also shows that SALT can make models more aligned with human preference without explicit reward models, which is the same with DPO. Also, it’s questionable whether a good explicit reward model can be learned from S_AI and S_E since it’s not as easy as positive/negative movie reviews to distinguish. For limitation, How does the SALT model generalize out of distribution, compared with PPO with an explicit reward function? For example, standard RLHF methods can leverage additional unlabeled prompts by labeling LM generations with the learned reward model. Can training with self-labeling from the SALT similarly make effective use of unlabeled prompts?
>
> 	> Other papers like RAFT and RRHF use explicit reward models to filter high-score data points for SFT. Whether we can train a good reward model as a good filter is also a big question here. Another difference is that, we will make full use of all data points (S_AI + S_E) during the training, but they will only use high quality one (S_E) and discard the rest (S_AI). So theoretically we use data more efficiently and model more information from S_AI.
>
> It’s amazing to see the RLHF community grow up so quickly! We will explore and try more experiments to compare SALT with all these new methods (DPO/RAFT/RRHF) in the future.
>
>
> **Q2:** Empirically, the improvements observed are marginal: for a baseline ROUGE of ~50, the improvement is ~0.5 (1%). The result possibly does not strongly encourage the adoption of the proposed method.
>
> **Answer:** Thanks to the reviewer's point here. We add 95% Confidence interval results here:
>
> |**Methods**|**CCUser_R1**|**CCUser_Uf**|**CC_R1**|**CC_Uf**|
> |-------|-------|-------|-------|-------|
> |**SALT(l)**|**57.77** (±0.28)|**61.02** (±1.06)|**34.27** (±0.21)|**46.45** (±0.52)|
> |**SALT(l_d)**|**57.70**  (±0.25) |**61.06** (±0.86)|**34.46**  (±0.31)|**46.58** (±0.33)|
> |**SALT(l_i)**|**57.84**  (±0.36)|**60.81** (±0.79)|**34.68** (±0.25)|**46.77** (±0.71)|
> |**SALT(u)**|**57.57** (±0.66)|**61.09** (±1.33)|**34.47** (±0.44)|**46.64**(±0.51)|
> |**SALT(l+u)**|**58.39** (±0.57)|**62.13** (±1.03)|**34.79** (±0.30)|**47.06** (±0.47)|
> |**SALT(l) +RSALT(l)**|**59.57** (±0.47)|**62.52** (±0.98)|**35.55** (±0.32)|**48.25** (±0.60)|
> |**SALT(l+u) +RSALT(l)**|**59.60** (±0.52)|**62.57** (±1.34)|**35.43** (±0.23)|**48.20** (±0.68)|
> |**SALT(l) +RSALT(l+u)**|**59.88** (±0.43)|**62.60** (±0.85)|**36.24** (±0.36)|**48.42** (±0.41)|
> |**SALT(l+u) +RSALT(l+u)**|**60.43** (±0.61)|**63.44** (±0.92)|**36.26** (±0.40)|**48.69** (±0.59)|
>
> In the appendix, **our human eval and gpt4 eval** also support the experimental results of SALT.
>
> **Q3:** Due to the way the data is processed, which uses humans, Google APIs, and the in-house model, it is hard to reproduce the work.
>
> **Answer:** Thanks to the reviewer's point here. Unfortunately, since CCUser is the product data, we can't release this patient-doctor-conversation-to-note data for reproduction due to the privacy restrictions.
>
> However, we know the importance of reproduction to the community, so we will release our SALT code, scores, and output results of different models on the public dataset. We couldn't find a big enough public dataset with (doctor-patient conversation, note) and human edits information to train a summarization model, so we experimented on the MIMIC-III after-visit-summary generation dataset [1] [2] for all SALT experiments. In addition, we use GPT4 to edit the model output to get "human edits". The dataset has 31,895 (Clinical note, AVS) pairs for this summarization task. We first fine-tune the gpt2 on this dataset, and then ask GPT4 to edit the trained-gpt2 generated summaries based on the input clinical notes and reference AVS summaries. GPT4 can easily provide such “human-edits” without too much prompt engineering. So we have (Clinical note, gpt2 generated summary, GPT4-edit summary) for SALT/DPO. Here are the results:
>
> |**Results with GPT2**|**Rewards ACC**|**Rouge1**|**Rouge2**|**Rougel**|**Meteor**
> |-------|-------|-------|-------|-------|-------|
> |**SALT(l)**|**0.596**|**0.500**|**0.366**|**0.487**|**0.455**|
> |**SALT(l_d)**|**0.586**|**0.512**|**0.378**|**0.498**|**0.471**|
> |**SALT(l_i)**|**0.677**|**0.502**|**0.369**|**0.489**|**0.457**|
> |**SALT(u)**|**0.860**|**0.462**|**0.317**|**0.447**|**0.406**|
> |**SALT(l+u)**|**0.777**|**0.511**|**0.375**|**0.497**|**0.468**|
> |**DPO(beta=0.1)**|**0.654**|**0.491**|**0.355**|**0.477**|**0.444**|
> |**DPO(beta=0.5)**|**0.827**|**0.485**|**0.347**|**0.470**|**0.438**|
>
> The results here are in line with paper results, we will release codes, results and model output on this public task for the community.
>
> [1] Generation of Patient After-Visit Summaries to Support Physicians
> https://aclanthology.org/2022.coling-1.544/
>
> [2] Learning to Revise References for Faithful Summarization
> https://aclanthology.org/2022.findings-emnlp.296.pdf
>
>
>
>
> ### Questions For The Authors:
> **Question** I find the data split not very clear. In 243-246, "we randomly select 70 notes from 7 physicians as training dataset, 10 for each physician, and divide the remaining 145 notes into 245 evaluation and test sets." Why is it important to mention the number of physicians covered in the training data? I also wonder if the training and test sets overlap in terms of being produced by the same physician. If so there might be implicit bias such as the use of words or the edit style.
>
>
> **Answer:** With regard to the number of doctors, we just report some information of our data truthfully. We try to make the sampling of data random. The data of train and test may come from the same doctor, but they will not come from the same note. We admit the existence of implicit bias, in fact this is a very interesting topic, we also mentioned this in the Limitation and Ethics Statement sections that “There will be some potential problems in actual deployment: First of all, for Human-Edits, we don't know whether the user is modifying because of some kind of error in S_AI or because of the user's personal preference. These two behaviors need to be distinguished in future research or actual deployment, because the former data is more suitable for improving the problems of the model itself (such as some actual errors), and the latter data is more suitable for user-personalized training data.”. We will keep exploring these directions in our future work.

---

### Official Review · Reviewer_6R2L · 2023-08-11

**Soundness:** 4

**Excitement:**

3: Ambivalent: It has merits (e.g., it reports state-of-the-art results, the idea is nice), but there are key weaknesses (e.g., it describes incremental work), and it can significantly benefit from another round of revision. However, I won't object to accepting it if my co-reviewers champion it.

**Missing References:**

[An Exploration of Post-Editing Effectiveness in Text Summarization](https://aclanthology.org/2022.naacl-main.35) (Lai et al., NAACL 2022)

**Paper Topic And Main Contributions:**

This paper proposed to use unlikelihood learning together with normal likelihood learning to incorporate human edits as feedbacks to improve text summarization.
The learning method use sequence alignment to identify the token changes the human make on AI-generated summary, such token changes are then encoded in the cross-entropy loss function to either penalize or promote their likelihood in the generation.
The experiments on clinical note generation (and other public data as presented in Appendix) show that unlikelihood learning with human edits can perform better than baseline that simple continual training on human edited data.
The experiments with imitation where human edits are simulated between reference ground truth summary and AI-generated summary, and confirm such data also help improve performance, meanwhile reduce the needs of costly human edits.
The experiments also show the unlikelihood learning with imitation edits can serve as a way of data augmentation to improve catastrophic forgetting of simply replay-based method.

**Questions For The Authors:**

Question A: The model training details are missing. How the base model M is trained, for how long, and what is the stop criteria? How do you make sure the model is fully trained without overfitting?

Question B: How does the unlikelihood learning compared to other learning method like RL and contrastive learning?

Question C: Does human edits always improve the quality of the summary in the dataset? it would be nice to show such statistics, on what edited, how the quality shift.

**Reasons To Accept:**

1. The proposed method is novel, by using unlikelihood learning to learn human edits as feedback to improve text summarization. The proposed method is generic and could be extended text generation in many other domains. The study also has a broad interest to the research in the community of human AI collaboration.
2. The study of unlikelihood with imitation edits raise interesting findings: (1) it can help further tuning the text generation model with less overfitting; (2) it could help eliminate forgetting issue when using replay-based method.
3. The experiments with careful analysis are clear and looks solid to prove the advantage of proposed method.

**Reasons To Reject:**

1. The paper does not compare to other methods mentioned in literature review on incorporate Human Edits as Feedback to improve text summarization, for example how the used unlikelihood learning method is compared to Reinforcement learning based and constrastive learning based.  The title seems to be a bit overselling.
2. The numbers shown in the experiment results look every close, some claims might not be true given the unclear significance. For example, in Table 3, the claims about comparing different variant of the proposed methods (SALT_x)
3. The evaluation majorly rely on automatic metric (ROUGE) to evaluate the summarization system, which has been proved to unreliable in many areas. The human evaluation is just a quick sanity check with very small (25) samples. It would be nice to expand this to larger one, and also show some case analysis on the model behaviors.
4. The paper mainly focus on single datasets, it would be nice to bring some experiments on other public dataset in the appendix to the main text.

**Reproducibility:**

4: Could mostly reproduce the results, but there may be some variation because of sample variance or minor variations in their interpretation of the protocol or method.

**Reviewer Confidence:**

4: Quite sure. I tried to check the important points carefully. It's unlikely, though conceivable, that I missed something that should affect my ratings.

---

> ### Author Rebuttal · Authors · 2023-08-29
>
> ### Reasons To Reject:
>
> **Q1:** The paper does not compare to other methods mentioned in literature review on incorporate Human Edits as Feedback to improve text summarization, for example how the used unlikelihood learning method is compared to Reinforcement learning based and constrastive learning based. The title seems to be a bit overselling.
>
> **Answer:** Thanks to the reviewer's point here. In order to solve this question, we experimented on the most representative work of RLHF recently, DPO (Direct Preference Optimization), on our CCUser dataset, because the settings of the two are similar, that is, training directly on the human preference dataset , without training explicit reward model. Specifically, we use S_AI as the rejected summary, S_E as the chosen summary and calculate the DPO loss to train the model.
>
> |**Results with GPT2**|**Rewards ACC**|**Rouge1**|**Rouge2**|**Rougel**|**Meteor**
> |-------|-------|-------|-------|-------|-------|
> |**SALT(l)**|**0.368**|**0.381**|**0.203**|**0.371**|**0.292**|
> |**SALT(l_d)**|**0.430**|**0.388**|**0.210**|**0.376**|**0.302**|
> |**SALT(l_i)**|**0.390**|**0.389**|**0.214**|**0.376**|**0.320**|
> |**SALT(u)**|**0.522**|**0.391**|**0.216**|**0.380**|**0.310**|
> |**SALT(l+u)**|**0.591**|**0.394**|**0.215**|**0.383**|**0.320**|
> |**DPO(beta=0.1)**|**0.484**|**0.379**|**0.210**|**0.369**|**0.301**|
> |**DPO(beta=0.5)**|**0.532**|**0.372**|**0.191**|**0.361**|**0.291**|
>
> Here are our findings:
>
> - It is difficult for DPO to learn an implicit reward through S_AI and S_E because most words in S_AI and S_E are the same. So we find the reward metrics (reward ACC or margin) and other metrics (ROUGE, Factuality score) become negatively correlated. If we change hyper-parameters to get high reward metrics, the ROUGE and Factuality score becomes extremely bad, and its final generated summary becomes a mess. On the contrary, if we want to keep a ROUGE, Factuality score, the reward metrics will not improve at all (similar to SFT training). It’s easy to explain from the DPO loss function:
>
> 	> pi_logratios = policy_chosen_logps - policy_rejected_logps
>
> 	> ref_logratios = reference_chosen_logps - reference_rejected_logps
>
> 	> logits = pi_logratios - ref_logratios
>
> 	> DPO loss = --F.logsigmoid(self.beta * logits)
>
> So DPO considered penalizing the whole rejected summary, which is not a suitable behavior with human edit feedback. Again, most words in S_AI and S_E are the same, so DPO does not consider such cases.
>
> - SALT can be treated as an extension or variance of DPO, especially for human edit feedback. One straightforward solution to improve DPO on human edit feedback is to target “negative tokens'' in the rejected summary and calculate DPO loss, which aligns with our SALT ideas. After doing sequence alignment to find negative tokens, we use likelihood and unlikelihood loss for different tokens, but DPO loss can also be a good choice. We also calculate DPO rewards when we train the model with SALT, and interestingly, we find that, even though SALT loss function doesn’t explicitly maximize chosen and rejected logps, we still see similar improvement for DPO rewards metrics (e.g., ACC or margin).
>
> 	> chosen_rewards = self.beta * (policy_chosen_logps - reference_chosen_logps)
>
> 	> rejected_rewards = self.beta * (policy_rejected_logps - reference_rejected_logps)
>
> We will update these small experiments in the appendix and discuss our findings. We also treat this as our future work with more deep and comprehensive experiments.
>
> - We don’t have enough time to try all recent RLHF methods, but here is some thoughts:
>
> 	> It seems that SALT keeps most of the advantages and disadvantages of DPO against PPO. Often, no reinforcement learning means more stable and easy training (and hyper-tuning). Our human eval also shows that SALT can make models more aligned with human preference without explicit reward models, which is the same with DPO. Also, it’s questionable whether a good explicit reward model can be learned from S_AI and S_E since it’s not as easy as positive/negative movie reviews to distinguish. For limitation, How does the SALT model generalize out of distribution, compared with PPO with an explicit reward function? For example, standard RLHF methods can leverage additional unlabeled prompts by labeling LM generations with the learned reward model. Can training with self-labeling from the SALT similarly make effective use of unlabeled prompts?
>
> 	> Other papers like RAFT and RRHF use explicit reward models to filter high-score data points for SFT. Whether we can train a good reward model as a good filter is also a big question here. Another difference is that, we will make full use of all data points (S_AI + S_E) during the training, but they will only use high quality one (S_E) and discard the rest (S_AI). So theoretically we use data more efficiently and model more information from S_AI.
>
> It’s amazing to see the RLHF community grow up so quickly! We will explore and try more experiments to compare SALT with all these new methods (DPO/RAFT/RRHF) in the future.
>
>
> **Q2:** The numbers shown in the experiment results look every close, some claims might not be true given the unclear significance. For example, in Table 3, the claims about comparing different variant of the proposed methods (SALT_x)
>
> **Answer:** 95% Confidence interval results:
>
> |**Methods**|**CCUser_R1**|**CCUser_Uf**|**CC_R1**|**CC_Uf**|
> |-------|-------|-------|-------|-------|
> |**SALT(l)**|**57.77** (±0.28)|**61.02** (±1.06)|**34.27** (±0.21)|**46.45** (±0.52)|
> |**SALT(l_d)**|**57.70**  (±0.25) |**61.06** (±0.86)|**34.46**  (±0.31)|**46.58** (±0.33)|
> |**SALT(l_i)**|**57.84**  (±0.36)|**60.81** (±0.79)|**34.68** (±0.25)|**46.77** (±0.71)|
> |**SALT(u)**|**57.57** (±0.66)|**61.09** (±1.33)|**34.47** (±0.44)|**46.64**(±0.51)|
> |**SALT(l+u)**|**58.39** (±0.57)|**62.13** (±1.03)|**34.79** (±0.30)|**47.06** (±0.47)|
> |**SALT(l) +RSALT(l)**|**59.57** (±0.47)|**62.52** (±0.98)|**35.55** (±0.32)|**48.25** (±0.60)|
> |**SALT(l+u) +RSALT(l)**|**59.60** (±0.52)|**62.57** (±1.34)|**35.43** (±0.23)|**48.20** (±0.68)|
> |**SALT(l) +RSALT(l+u)**|**59.88** (±0.43)|**62.60** (±0.85)|**36.24** (±0.36)|**48.42** (±0.41)|
> |**SALT(l+u) +RSALT(l+u)**|**60.43** (±0.61)|**63.44** (±0.92)|**36.26** (±0.40)|**48.69** (±0.59)|
>
>
> **Q3:** The evaluation majorly rely on automatic metric (ROUGE) to evaluate the summarization system, which has been proved to unreliable in many areas. The human evaluation is just a quick sanity check with very small (25) samples. It would be nice to expand this to larger one, and also show some case analysis on the model behaviors.
>
> **Answer:** Thanks to the reviewer for this suggestion. Unfortunately, since CCUser is the product data, we can't even do human eval due to the privacy restrictions, that's why we only do human eval with CC (human annotated data). But we did GPT4 eval on both CC and CCUser, and the results were as expected. At the same time, due to data privacy restrictions, we cannot provide specific cases of different behavior after SALT training on CCUser in the paper.
>
> **However**, we know the importance of reproduction and these case studies to the community, so we will release our SALT code, scores, and output results of different models on the public dataset. We couldn't find a big enough public dataset with (doctor-patient conversation, note) and human edits information to train a summarization model, so we used another clinical summarization task on MIMIC-III, after-visit-summary generation task for our public dataset experiments. In addition, we use GPT4 to edit the model output to get "human edits". The specific experimental results are given in the next point.
>
>
> **Q2:** The paper mainly focus on single datasets, it would be nice to bring some experiments on other public dataset in the appendix to the main text.
>
> **Answer:** Thanks to the reviewer's point here. In order to solve this question, we experimented on the MIMIC-III after-visit-summary generation dataset [1] [2]. The dataset has 31,895 (Clinical note, AVS) pairs for this summarization task. We first fine-tune the gpt2 on this dataset, and then ask GPT4 to edit the trained-gpt2 generated summaries based on the input clinical notes and reference AVS summaries. GPT4 can easily provide such “human-edits” without too much prompt engineering. So we have (Clinical note, gpt2 generated summary, GPT4-edit summary) for SALT/DPO. Here are the results:
>
> |**Results with GPT2**|**Rewards ACC**|**Rouge1**|**Rouge2**|**Rougel**|**Meteor**
> |-------|-------|-------|-------|-------|-------|
> |**SALT(l)**|**0.596**|**0.500**|**0.366**|**0.487**|**0.455**|
> |**SALT(l_d)**|**0.586**|**0.512**|**0.378**|**0.498**|**0.471**|
> |**SALT(l_i)**|**0.677**|**0.502**|**0.369**|**0.489**|**0.457**|
> |**SALT(u)**|**0.860**|**0.462**|**0.317**|**0.447**|**0.406**|
> |**SALT(l+u)**|**0.777**|**0.511**|**0.375**|**0.497**|**0.468**|
> |**DPO(beta=0.1)**|**0.654**|**0.491**|**0.355**|**0.477**|**0.444**|
> |**DPO(beta=0.5)**|**0.827**|**0.485**|**0.347**|**0.470**|**0.438**|
>
> The results here are in line with paper results, we will release codes, results and model output on this public task for the community.
>
> [1] Generation of Patient After-Visit Summaries to Support Physicians
> https://aclanthology.org/2022.coling-1.544/
>
> [2] Learning to Revise References for Faithful Summarization
> https://aclanthology.org/2022.findings-emnlp.296.pdf
>
> ### Questions For The Authors:
> **Question A:** The model training details are missing. How the base model M is trained, for how long, and what is the stop criteria? How do you make sure the model is fully trained without overfitting?
>
> **Answer:** We talk about these in the Appendix (A.2 Implementation Details) to save some space. We trained the model for 10000 steps (about 10 epochs), we saved the model according to the best 3 rouge1 score, finally we use the epoch 8 checkpoint since it has the best rogue 1 score.
>
> **Question B:** How does the unlikelihood learning compared to other learning method like RL and contrastive learning?
>
> **Answer:** The answer here is the same with point1 in Reasons To Reject.
>
> **Question C:** Does human edits always improve the quality of the summary in the dataset? it would be nice to show such statistics, on what edited, how the quality shift.
>
> **Answer:** This is a very good question and future research topic, we also mentioned this in the Limitation and Ethics Statement sections that “There will be some potential problems in actual deployment: First of all, for Human-Edits, we don't know whether the user is modifying because of some kind of error in S_AI or because of the user's personal preference. These two behaviors need to be distinguished in future research or actual deployment, because the former data is more suitable for improving the problems of the model itself (such as some factual errors), and the latter data is more suitable for user-personalized training data.”. We will keep explore these directions in our future work.

---

### Meta-Review · Area_Chair_TKZr · 2023-09-17

**Recommendation:** 5

**Metareview:**

The paper introduces SALT, a novel method utilizing unlikelihood training with human edits to enhance clinical text summarization. Reviewers find the concept of incorporating human edits well-motivated and appreciate the comprehensive experiments and ablations. However, concerns about presentation clarity in certain sections and the effectiveness of imitation edits are raised. Reviewers suggest broader comparisons with prior work, the adoption of established edit-based metrics, and a deeper exploration of alignment between AI-generated summaries and ground truth.

---

### Decision · Program_Chairs · 2023-10-07

**Decision:**

Accept-Main

**Comment:**

The paper introduces SALT, a novel method utilizing unlikelihood training with human edits to enhance clinical text summarization. Reviewers find the concept of incorporating human edits well-motivated and appreciate the comprehensive experiments and ablations. However, concerns about presentation clarity in certain sections and the effectiveness of imitation edits are raised. Reviewers suggest broader comparisons with prior work, the adoption of established edit-based metrics, and a deeper exploration of alignment between AI-generated summaries and ground truth.